# Hermite-NGP: Gradient-Augmented Hash Encoding for Learning PDEs

Jinjin He [1]   Zhiqi Li [1]   Sinan Wang [1]   Bo Zhu [1]

## Abstract

We propose *Hermite-NGP*, a gradient-augmented multi-resolution hash encoding designed to enable fast and accurate computation of spatial derivatives for neural PDE solvers. Unlike existing NGP-based approaches that rely on automatic differentiation or finite differences and suffer from instability or high cost, Hermite-NGP explicitly stores function values and mixed partial derivatives at hash grid vertices, allowing fully analytic evaluation of gradients, Jacobians, and Hessians via Hermite interpolation. This design preserves the efficiency and spatial adaptivity of NGP while supporting analytic differential operators up to second order. We further introduce a multi-resolution curriculum training strategy analogous to multigrid V-cycles to enable coarse-to-fine optimization. Across a range of 2D and 3D PDE benchmarks, Hermite-NGP achieves up to $\sim 20\times$ lower error than prior neural PDE methods, and reduces wall-clock convergence time by $2$–$10\times$ compared to other solvers, with per-epoch training times as low as $3.5\,\mathrm{ms}$ for models with up to 17M parameters.

## 1. Introduction

Efficient neural scene representations that combine classical data structures (grids, tensors, Gaussians) with neural counterparts have emerged as a highly effective family of spatial representations for neural field learning, offering strong locality, spatial adaptivity, and instant training in high-dimensional function approximation. Multi-resolution hash tables, popularized by Instant Neural Graphics Primitives (I-NGP) (Müller et al., 2022), are one such design; related approaches include grid- and tensor-decomposition representations (Fridovich-Keil et al., 2022; 2023; Chen et al., 2022; Cao & Johnson, 2023; Kim et al., 2024; Chen

[1]Georgia Institute of Technology, Atlanta, GA, USA. Correspondence to: Jinjin He <jhe433@gatech.edu>.

*Proceedings of the 43rd International Conference on Machine Learning*, Seoul, South Korea. PMLR 306, 2026. Copyright 2026 by the author(s).

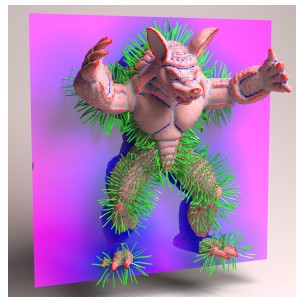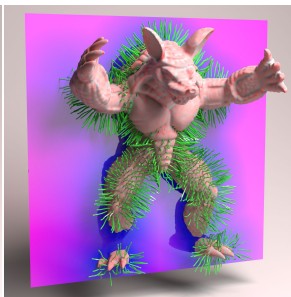

*Figure 1.* **SDF Gradients and Curvature on Armadillo.** Our method (left) recovers smoother gradient and curvature fields than NeuralAngelo (Li et al., 2023b) (right), used here as an FD-based hash-encoding baseline under the same SDF+gradient objective. Gradients are visualized as green line segments around the surface, while curvature is shown via mesh coloring.

et al., 2025; Zou et al., 2024), anti-aliased NeRF variants (Barron et al., 2023), and Gaussian-based methods such as 3DGS (Kerbl et al., 2023). These computational merits make NGP-style encodings particularly attractive for representation learning problems that demand both accuracy and efficiency, such as neural radiance field, signed distance field, image reconstruction, and so on (Chen et al., 2023a; Hu et al., 2023; Chen et al., 2023b; Li et al., 2023b; Yu et al., 2022; Wang et al., 2023b; Müller et al., 2022).

However, despite their success in representing graphics primitives, NGP-style encodings prove inadequate for learning PDE-governed functions, especially in the context of physics-informed neural networks (PINNs) that rely on accurate differential constraints (Raissi et al., 2020; Karniadakis et al., 2021; Raissi et al., 2019; 2017). *The core challenge lies in I-NGP's inherent inability to compute spatial derivatives efficiently*: standard hash encodings do not support evaluation of first- and second-order derivatives via automatic differentiation, while finite-difference approximations incur prohibitive computational cost and are highly sensitive to the choice of discretization parameters. As a result, naive applications of NGP to neural PDE settings often face challenges in addressing differentiation (see (Huang & Alkhalifah, 2024; Li et al., 2023b; Chetan et al., 2025; Wang et al., 2024a) for examples). This mismatch constitutes a fundamental barrier to leveraging the efficiency of NGP-style representations for solving PDE learning problems.

To address this challenge of efficient differentiation, we pro-

pose *Hermite-NGP*, a gradient-augmented multi-resolution hash encoding designed to support analytic derivative evaluation within a neural representation. Our key insight is inspired by the gradient-augmented field representations (e.g., gradient-augmented level set (Nave et al., 2010), affine particle-in-cell (Jiang et al., 2015), and flow map methods (Zhou et al., 2024; Deng et al., 2023), where a scalar field discretized on a grid is augmented with an auxiliary, collocated gradient field, enabling continuous and stable derivative reconstruction via Hermite interpolation within each grid cell (see Fig. 3). Rather than approximating derivatives using finite-difference stencils or recovering them implicitly through automatic differentiation, these methods treat derivative information as a first-class component of the representation itself. By explicitly parameterizing and evolving the gradient field alongside the primary scalar field, gradient-augmented representations yield well-defined, easy-to-calculate differential operators throughout the domain. Such representations have been widely adopted in computational physics and later computer graphics for various simulation problems involving dynamic interfaces (Li et al., 2023a; Kolomenskiy et al., 2016; Kolahdouz & Salac, 2013), material transport (Anumolu & Trujillo, 2018; Lee et al., 2014), and vortical structures (Bøckmann & Vartdal, 2014; Mercier et al., 2020; Zhou et al., 2024).

Hermite-NGP explicitly parameterizes both the hashed feature function and its mixed partial derivatives at each hash grid point. For each spatial resolution, we store Hermite interpolation coefficients corresponding to the function value and its partial derivatives and jointly optimize these quantities during training, resulting in a coupled representation that directly encodes local differential structure. Given these learned coefficients across hash resolutions, Hermite interpolation reconstructs a globally $C^1$-continuous spatial field whose second-order derivatives are analytic within each cell (piecewise across cell boundaries). This representation-level design eliminates truncation error from numerical differencing and avoids instability caused by automatic differentiation through discontinuous hash encodings, enabling analytic evaluation of gradients, Jacobians, Hessians, and Laplacians in a single forward pass while preserving the locality, adaptivity, and scalability of NGP-style encodings (see Figure 1 for an example). We demonstrate the efficacy of our approach across diverse 2D and 3D PDE benchmarks by achieving relative $L^2$ errors down to $10^{-5}$, offering up to $10\times$ accuracy gains over state-of-the-art neural PDE solvers and up to $10^3\times$ improvement over baseline NGP methods, while converging within minutes with per-epoch training costs of $2$–$3.5\,\mathrm{ms}$ on a single GPU.

Our main contributions are:

- **Gradient-Augmented Hash Encoding**. We propose a gradient-augmented hash encoding based on Hermite interpolation that stores mixed partial derivatives at grid vertices, enabling $C^1$ Hermite interpolation with non-trivial second-order structure.
- **Analytic Derivative Evaluation**. The proposed encoding admits fully analytic computation of first- and second-order spatial derivatives, avoiding finite-difference approximations and auto-differentiation.
- **Multi-resolution Coarse-to-Fine Training**. We develop a coarse-to-fine training strategy that leverages the hierarchical structure of Hermite-NGP, inspired by multigrid methods for PDE optimization.
- **Neural PDE Solving with Complex Geometry**. We demonstrate Hermite-NGP across diverse 2D/3D PDEs, complex geometric domains, and intrinsic differential operators within a unified learning framework.

## 2. Related Work

**Physics-Informed Neural Networks.** PINNs (Raissi et al., 2017; 2019) embed PDE constraints into networks to obtain mesh-free solutions (Karniadakis et al., 2021) with applications in fluid mechanics (Raissi et al., 2020) and libraries like DeepXDE (Lu et al., 2021b). Key challenges include spectral bias (Rahaman et al., 2019), gradient pathologies (Wang et al., 2021), and loss imbalances (Wang et al., 2022), addressed by adaptive weighting (McClenny & Braga-Neto, 2023), causal training (Wang et al., 2024c), curriculum strategies (Krishnapriyan et al., 2021; Duan et al.), complex geometry handling (Costabal et al., 2024), and high-dimensional techniques (Hu et al., 2024); recent benchmarks (Zhongkai et al., 2024) evaluate these systematically. Alternative representations include Fourier features (Tancik et al., 2020), PIXEL (Kang et al., 2023), SPINN (Cho et al., 2023), and PIG (Kang et al., 2024). Neural operators (Lu et al., 2021a; Li et al., 2021) learn solution mappings but require supervision.

**Multi-Resolution Hash Encoding.** Instant NGP (Müller et al., 2022) introduced compact multi-resolution hash tables with $O(1)$ lookup for neural radiance fields (Mildenhall et al., 2021), inspiring extensions such as Zip-NeRF (Barron et al., 2023), Neuralangelo (Li et al., 2023b), NeuS2 (Wang et al., 2023b), and Nerfstudio (Tancik et al., 2023). Related grid- and tensor-decomposition representations include TensoRF (Chen et al., 2022); Gaussian-based methods such as 3D Gaussian Splatting (Kerbl et al., 2023) occupy a distinct but adjacent design space. Recent work also applies hash encodings to efficient PINN training (Huang & Alkhalifah, 2024; Wang et al., 2024a).

**Gradient-Augmented Representation** In level set methods, gradient-augmented (Nave et al., 2010; Bøckmann & Vartdal, 2014) approaches improve interface tracking accuracy by maintaining both signed distance and its gradient (Jiang et al., 2015). Similar ideas appear in fluid simula-

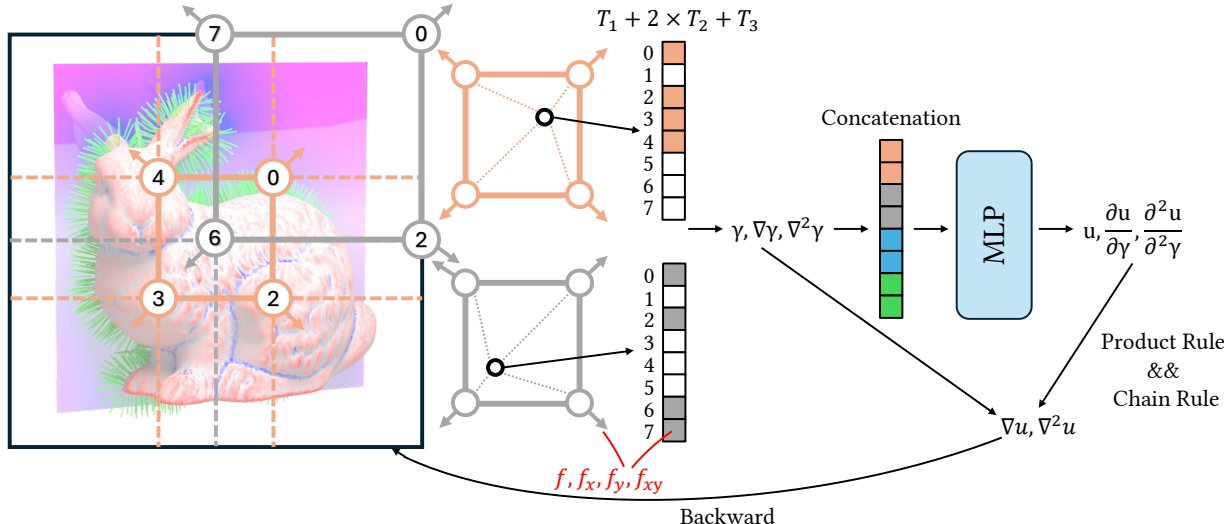

*Figure 2.* **Hermite-NGP Workflow.** Multi-resolution grids store Hermite coefficients $(f, f_x, f_y, f_{xy})$ in separate hash tables with sizes $T_1$ (values), $T_2$ (first derivatives), and $T_3$ (mixed derivatives), producing encoding $\gamma, \nabla\gamma, \nabla^2\gamma$. The MLP outputs $u, \partial u/\partial\gamma, \partial^2 u/\partial\gamma^2$, combined via chain rule to yield $\nabla u, \nabla^2 u$ for PDE loss.

tion, where velocity gradients enable higher-order advection schemes. Recent neural representations have explored gradient supervision for improved surface reconstruction and normal estimation (Sommer et al., 2022; Huang et al., 2024).

## 3. Background

**Multi-Resolution Hash Encoding** Multi-resolution hash encoding (Müller et al., 2022) represents a function using $L$ resolution levels of learnable hash tables, each storing $F$-dimensional features. At level $l \in \{0, \ldots, L-1\}$, an implicit $d$-dimensional grid has resolution $N_l = \lfloor N_{\min} b^l \rfloor$, where $b = \exp\left(\frac{\ln N_{\max} - \ln N_{\min}}{L-1}\right)$.

Given a query point $\mathbf{x} \in \mathbb{R}^d$, the $2^d$ neighboring grid vertices $\{g^{i,l}\}$ are identified. Their features are retrieved via the hash function $h(g^{i,l}) = \left(\bigoplus_{j=0}^{d-1} g_j^{i,l} \pi_j\right) \mod T^l$, where $\bigoplus$ denotes XOR, $\{\pi_j\}$ are fixed primes, and $T^l$ is the table size. $d$-linear interpolation over these features yields the encoding $\gamma^l(\mathbf{x})$. Encodings from all levels are concatenated as $\gamma(\mathbf{x}) \in \mathbb{R}^{L \cdot F}$ and fed to a lightweight MLP. However, $d$-linear interpolation is only $C^0$ continuous: first-order derivatives are piecewise constant and discontinuous across cell boundaries, rendering higher-order derivatives undefined. This precludes analytic evaluation of PDE operators involving first- or second-order derivatives (first derivatives jump across cell boundaries; second derivatives vanish within cells).

**Physics-Informed Neural Networks** Physics-Informed Neural Networks (PINNs) (Raissi et al., 2019) solve PDEs by embedding physical constraints into neural network training. Given a PDE $\mathcal{N}_{x,t}[u](x,t) = f(x,t)$ on domain

$\Omega \times [0, T]$ with initial condition $u(x, 0) = g(x)$ and boundary condition $\mathcal{B}_{x,t}[u] = h(x,t)$ on $\partial\Omega$, PINNs approximate $u$ using a neural network $u_\theta(x, t)$ by minimizing

$$\mathcal{L}(\theta) = \lambda_{\text{res}}\mathcal{L}_{\text{res}} + \lambda_{\text{ic}}\mathcal{L}_{\text{ic}} + \lambda_{\text{bc}}\mathcal{L}_{\text{bc}} + \lambda_{\text{data}}\mathcal{L}_{\text{data}}, \quad (1)$$

where $\mathcal{L}_{\text{res}}$ enforces the PDE residual at collocation points, and the remaining terms impose initial, boundary, and data constraints. Derivative boundary conditions (e.g., Neumann) are imposed identically, via a soft loss applied to the corresponding boundary collocation set.

## 4. Gradient-Augmented Representation

**Naming Convention.** We denote the spatial dimension by $d$, the number of resolution levels by $L$, grid resolution at level $l$ by $N_l$, and grid spacing by $\Delta x_l$. Hash table sizes for derivative type $i$ at level $l$ are $T_i^l$, with feature dimension $F$ per stored value. Multi-index $\alpha \in \{0, 1\}^d$ specifies partial derivative orders, $\theta_{l,k}^{(\alpha)}$ denotes Hermite coefficients at level $l$ and hash index $k$, and $H^{(\alpha)}$ the corresponding Hermite basis. The SIREN frequency parameter is $\omega$.

### 4.1. Hermite Interpolation

Hermite interpolation constructs smooth approximations using both function values and derivatives at grid points (Hermite & Borchardt, 1878; Birkhoff et al., 1968; Ciarlet & Raviart, 1972). In 1D, given function values $f_0, f_1$ and derivatives $f_0', f_1'$ at endpoints of interval $[0, 1]$, the cubic

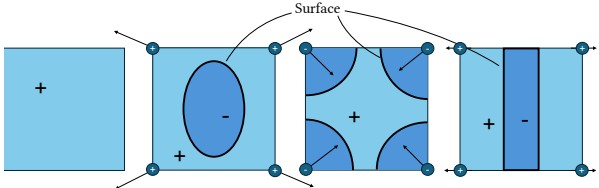

*Figure 3.* Gradient-augmented illustration. The left cell without gradients yields constant interpolation, while the others use gradients to produce rich sub-grid features.

Hermite interpolant is:

$$H(t) = f_0 h^{(0)}(t) + f_1 h^{(0)}(1-t) + f_0' h^{(1)}(t) + f_1' h^{(1)}(t),$$

$$h^{(0)}(t) = \begin{cases} -2t^3 + 3t^2, & 0 \le t \le 1, \\ 0, & \text{otherwise,} \end{cases}$$

$$h^{(1)}(t) = \begin{cases} 2t^3 - 3t^2 + 1, & 0 \le t \le 1, \\ 0, & \text{otherwise,} \end{cases}$$

$$(2)$$

where $h^{(i)}$, $i \in \{0, 1\}$, denote the basis functions.

For a scalar field $f(\mathbf{x})$ in $d$ dimensions defined on a grid with spacing $\Delta x$, at each grid vertex $g$ located at position $\mathbf{x}_g$, we store $2^d$ values, including the function value $f_g = f(\mathbf{x}_g)$ and the mixed partial derivatives $f_g^{(\alpha)} = \frac{\partial^{|\alpha|} f}{(\partial \mathbf{x})_\alpha}|_{\mathbf{x}=\mathbf{x}_g}, \alpha \in \{0,1\}^d$, where $\alpha = (\alpha_i)_{i=0}^{d-1}$ denotes a multi-index and $|\alpha| = \sum_i \alpha_i$. For example, in three dimensions, $f_g^{(0,1,1)} = \frac{\partial^2 f}{\partial x_2 \partial x_3}|_{\mathbf{x}=\mathbf{x}_g}$. Then the Hermite interpolant is constructed via tensor products:

$$H[f](\mathbf{x}) = \sum_{g \in N(\mathbf{x})} \sum_{\alpha \in \{0,1\}^d} f_g^{(\alpha)} H^{(\alpha)}\left(\frac{\mathbf{x} - \mathbf{x}_g}{\Delta x}\right) \Delta x^{|\alpha|},$$

$$(3)$$

where $N(\mathbf{x})$ denotes the $2^d$ vertices of the cell containing $\mathbf{x}$, $H^{(\alpha)}(\mathbf{x}) = \Pi_{i=1}^d h^{(\alpha_i)}(x_i)$ are basis functions for $d$ dimensions. This construction guarantees $C^1$ continuity with gradients exactly recovered at grid points: $\nabla H[f](\mathbf{x}_g) = (f_g^{(1,0,\dots)}, f_g^{(0,1,\dots)}, \dots)$. Unlike $d$-linear interpolation where $\nabla^2 u \equiv 0$ inside cells, Hermite interpolation provides well-defined, non-zero second derivatives throughout the domain.

### 4.2. Hermite Hash Encoding

Standard hash encoding stores only function values at grid vertices and relies on $d$-linear interpolation. While simple and efficient, this yields only $C^0$ continuity: first derivatives are piecewise constant (giving $\nabla^2 u \equiv 0$ inside cells) and discontinuous at cell boundaries. For PINNs, this fundamentally prevents analytic computation of higher-order derivatives like the Laplacian. Our key insight is that by storing the complete set of mixed partial derivatives $\partial^\alpha f$ for $\alpha \in \{0, 1\}^d$ at each vertex, we have exactly the degrees of freedom needed for $C^1$ Hermite interpolation (Ciarlet & Raviart, 1972) with well-defined second derivatives.

For each hash entry, we store the complete set of partial derivatives $\{f^{(\alpha)}\}_{\alpha \in \{0,1\}^d}$—e.g., $(f, f_x, f_y, f_{xy})$ in 2D—requiring $2^d$ coefficients per vertex rather than one. To manage this increased storage efficiently, we maintain *separate hash tables* grouped by derivative type. In 2D, this yields three tables: (1) $T_1 \times F$ for function values $f$; (2) $T_2 \times 2F$ for first derivatives $(f_x, f_y)$; and (3) $T_3 \times F$ for the mixed derivative $f_{xy}$. This separation allows different tables to use different sizes based on their representational requirements.

At resolution level $l$ with grid spacing $\Delta x_l = 1/N_l$, the Hermite hash encoding is:

$$\gamma^l(\mathbf{x}) = \sum_{g \in N_l(\mathbf{x})} \sum_{\alpha \in \{0,1\}^d} \theta_{l,h(g)}^{(\alpha)} H^{(\alpha)}\left(\frac{\mathbf{x} - \mathbf{x}_g}{\Delta x_l}\right) \Delta x_l^{|\alpha|},$$

$$(4)$$

where $N_l(\mathbf{x})$ denotes the $2^d$ vertices of the cell containing $\mathbf{x}$ at level $l$, $\theta_{l,h(g)}^{(\alpha)}$ are the learnable Hermite coefficients retrieved from the hash table corresponding to derivative order $|\alpha|$, and $H^{(\alpha)}$ are the basis functions defined in Section 3. Like standard NGP, hash collisions within each table are resolved implicitly through gradient-based optimization.

Features from all $L$ resolution levels are concatenated to form the final encoding $\gamma(\mathbf{x}) = (\gamma^0(\mathbf{x}), \dots, \gamma^{L-1}(\mathbf{x})) \in \mathbb{R}^{L \cdot F}$, where $F$ is the feature dimension per stored value. In 2D, the total parameter count across all tables is $L \times (T_1 + 2T_2 + T_3) \times F$. When all tables use the same size $T$, this simplifies to $L \times 4T \times F$, matching the $2^d$ factor from the 4 Hermite coefficients per vertex.

A key advantage of storing derivative information explicitly is that the encoding captures richer local spatial structure than function values alone. This enables the network to represent sharp gradients and rapid spatial variations directly in the encoding, which is particularly beneficial for PDEs with complex boundary conditions—the derivative coefficients can adapt locally to enforce boundary constraints while the function values capture the global solution structure. This spatial adaptivity, inherited from hash encoding's irregular grid structure, allows Hermite-NGP to handle complex geometric boundaries naturally without requiring boundary-conforming meshes. Note that the hash function $h(\cdot)$ is a discrete index lookup mapping integer grid coordinates to table entries: it is not part of the continuous computation graph, and all spatial derivatives are computed through the smooth Hermite basis applied to the retrieved coefficients.

## 5. Analytic Differentiation

### 5.1. Analytic Derivative Calculation

PDE residuals like the Laplacian $\nabla^2 u$ require second-order spatial derivatives. Standard autodifferentiation cannot provide meaningful second derivatives for hash encoding: the

$d$-linear interpolation yields piecewise constant first derivatives, making second derivatives zero almost everywhere. Standard NGP therefore resorts to finite differences (FD) like INGP-FD (Huang & Alkhalifah, 2024; Li et al., 2023b; Wang et al., 2024a), requiring $2d + 1$ forward passes (5 in 2D, 7 in 3D) and introducing $O(\epsilon^2)$ truncation error in the estimated second derivatives of the model output. This truncation error creates a fundamental accuracy ceiling around $10^{-5}$, regardless of how long training continues. For applications requiring higher precision, analytic derivatives are therefore essential rather than merely preferable. Hermite encoding enables analytic computation in a single forward pass by differentiating the basis functions directly.

Derivatives of the Hermite encoding follow by differentiating the basis functions. Defining $\mathbf{t} = (\mathbf{x} - \mathbf{x}_g)/\Delta x_l$, we have:

$$\partial_{i_1 \cdots i_k}^k \gamma_l = \sum_{g \in N_l(\mathbf{x})} \sum_{\alpha \in \{0,1\}^d} \theta_{l,h(g)}^{(\alpha)} \Delta x_l^{|\alpha|-k} \partial_{i_1 \cdots i_k}^k H^{(\alpha)}(\mathbf{t}). \tag{5}$$

For first derivatives ($k = 1$) and second derivatives ($k = 2$, enabling analytic Laplacian), the scaling $\Delta x_l^{|\alpha|-k}$ ensures correct dimensional behavior across resolution levels.

These formulas provide analytic derivatives of the Hermite interpolant within each cell, free of truncation error.

For implementation, the derivatives of the 1D Hermite basis functions are:

$$\frac{\partial h^{(0)}}{\partial t}(t) = \begin{cases} -6t^2 + 6t, & 0 \le t < 1, \\ 6t^2 + 6t, & -1 < t < 0, \\ 0, & |t| > 1, \end{cases} \quad \frac{\partial^2 h^{(0)}}{\partial t^2}(t) = \begin{cases} -12t + 6, & 0 \le t < 1, \\ 12t + 6, & -1 < t < 0, \\ 0, & |t| \ge 1, \end{cases}$$

$$\frac{\partial h^{(1)}}{\partial t}(t) = \begin{cases} 3t^2 - 4t + 1, & |t| < 1, \\ 0, & |t| \ge 1, \end{cases} \quad \frac{\partial^2 h^{(1)}}{\partial t^2}(t) = \begin{cases} 6t - 4, & |t| < 1, \\ 0, & |t| \ge 1, \end{cases} \tag{6}$$

For the $d$-dimensional Hermite basis, derivatives can be factorized as $\frac{\partial H^{(\alpha)}(\mathbf{y})}{\partial x_i} = \frac{\partial h^{(i)}(y_i)}{\partial x_i} \Pi_{k \neq i} h^{(k)}(y_k)$, $\frac{\partial^2 H^{(\alpha)}(\mathbf{y})}{\partial x_i^2} = \frac{\partial^2 h^{(i)}(y_i)}{\partial x_i^2} \Pi_{k \neq i} h^{(k)}(y_k)$ and $\frac{\partial^2 H^{(\alpha)}(\mathbf{y})}{\partial x_i \partial x_j} = \frac{\partial h^{(i)}(y_i)}{\partial x_i} \frac{\partial h^{(j)}(y_j)}{\partial x_j} \Pi_{k \neq i,j} h^{(k)}(y_k)$ for $i \neq j$, where $\mathbf{y} = (\frac{\mathbf{x} - \mathbf{x}_g}{\Delta x_l})$ (see Appendix A.1 for the complete 2D worked example). This factorization enables efficient vectorized computation: we compute 1D basis values and derivatives once per dimension, then combine them via outer products. The Laplacian is computed analytically by summing diagonal Hessian entries: $\nabla^2 \gamma = \sum_{i=1}^d \frac{\partial^2 \gamma}{\partial x_i^2}$.

Furthermore, our grouped table structure (Section 4.2) allows allocating smaller tables to higher-order derivatives, providing fine-grained control over the memory-accuracy tradeoff(see Section 7.7 for ablation study).

**Algorithm 1** Hermite-NGP Training

---

**Require:** Points $\mathbf{x} \in \mathbb{R}^{N \times d}$, hash tables $\{\theta_l^{(\alpha)}\}$, MLP $\phi$, active levels $L_{\text{active}}$
**Ensure:** PDE solution $u(\mathbf{x})$, derivatives $\nabla u, \nabla^2 u$
 1: **// Hermite Hash Encoding (Sec. 4.2)**
 2: **for** $l = 1, \ldots, L_{\text{active}}$ **do**          ▷ Coarse-to-fine: Sec. 6
 3:      $g \leftarrow N_l(\mathbf{x})$                    ▷ $2^d$ grid vertices
 4:      $k \leftarrow h(g) \mod T_l$                  ▷ Hash function
 5:      $\theta_{l,k}^{(\alpha)} \leftarrow \text{Lookup}(\theta_l, k)$ for $\alpha \in \{0,1\}^d$
 6:      $\gamma_l, \nabla \gamma_l, \nabla^2 \gamma_l \leftarrow \sum_g \sum_\alpha \theta_{l,k}^{(\alpha)} H^{(\alpha)}(\mathbf{x})$      ▷ Eq. 4
 7: **end for**
 8: $\gamma \leftarrow [\gamma_1; \ldots; \gamma_{L_{\text{active}}}]$          ▷ Concat multi-resolution
 9: **// MLP with Analytic Derivatives (Sec. 5.2)**
10: $u, \frac{\partial u}{\partial \gamma}, \frac{\partial^2 u}{\partial \gamma^2} \leftarrow \text{MLP}_\phi(\gamma)$                  ▷ SIREN
11: $\nabla u \leftarrow \frac{\partial u}{\partial \gamma} \nabla \gamma; \quad \nabla^2 u \leftarrow \frac{\partial^2 u}{\partial \gamma^2}(\nabla \gamma)^2 + \frac{\partial u}{\partial \gamma} \nabla^2 \gamma$
12: **// PDE Loss and Backward**
13: $\mathcal{L} \leftarrow \mathcal{L}_{\text{pde}}(u, \nabla u, \nabla^2 u) + \lambda_{\text{bc}} \mathcal{L}_{\text{bc}}$
14: $\text{Backward}(\mathcal{L}) \rightarrow$ update $\{\theta_l^{(\alpha)}\}, \phi$
15: **return** $u, \nabla u, \nabla^2 u$

---

### 5.2. End-to-End Differentiation

The complete Hermite-NGP model computes $u_\theta(\mathbf{x}) = \text{MLP}_\phi(\gamma(\mathbf{x}))$, where $\gamma(\mathbf{x})$ is the Hermite hash encoding. A key advantage is that analytic derivatives from the encoding propagate through the MLP to obtain differential operators without finite differences or autograd overhead.

We use SIREN activations (Sitzmann et al., 2020) with $\sigma(x) = \sin(\omega x)$, satisfying $\sigma'' = -\omega^2 \sigma$. For a single layer $u = W_2 \sin(\omega(W_1 \gamma + b_1)) + b_2$, denoting $z = W_1 \gamma + b_1$ and $a = \sin(\omega z)$, the Laplacian is:

$$\nabla^2 u = W_2 \left[ -\omega^2 a \odot \sum_{i=1}^d (W_1 \gamma_{x_i})^2 + \omega \cos(\omega z) \odot W_1 \nabla^2 \gamma \right], \tag{7}$$

where $\gamma_{x_i} = \frac{\partial \gamma}{\partial x_i}$ and $\nabla^2 \gamma$ are the analytic encoding derivatives from Section 5.1. All terms except $\gamma_{x_i}$ and $\nabla^2 \gamma$ reuse quantities from the forward pass.

For $K$-layer networks, derivatives propagate recursively. Let $a^{(0)} = \gamma$ and $a^{(k)} = \sin(\omega z^{(k)})$ where $z^{(k)} = W_k a^{(k-1)} + b_k$:

$$\frac{\partial a^{(k)}}{\partial x_i} = \omega \cos(\omega z^{(k)}) \odot W_k \frac{\partial a^{(k-1)}}{\partial x_i}, \tag{8}$$

$$\frac{\partial^2 a^{(k)}}{\partial x_i^2} = -\omega^2 a^{(k)} \odot \left( W_k \frac{\partial a^{(k-1)}}{\partial x_i} \right)^2 \tag{9}$$

$$+ \omega \cos(\omega z^{(k)}) \odot W_k \frac{\partial^2 a^{(k-1)}}{\partial x_i^2}. \tag{10}$$

The base case uses Hermite encoding derivatives; the full Laplacian $\nabla^2 u = \sum_{i=1}^d \frac{\partial^2 u}{\partial x_i^2}$ is computed in a single forward pass (see Appendix A.2). We initialize MLP weights

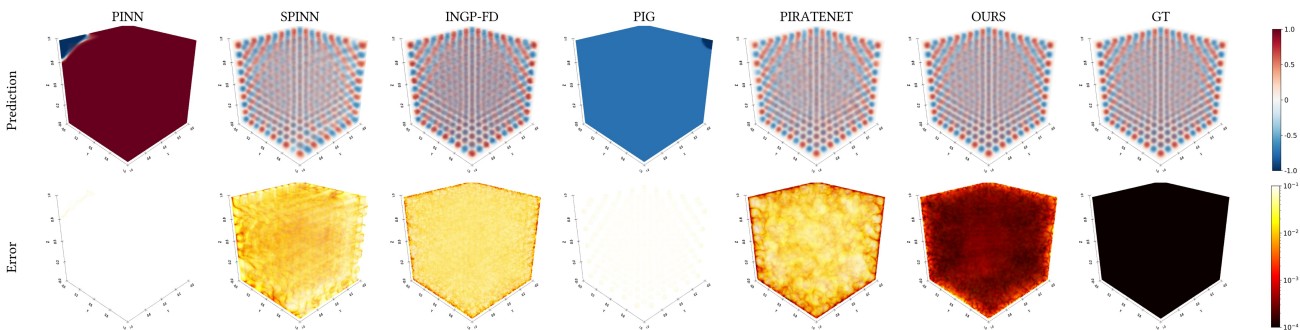

*Figure 4.* **Helmholtz 3D** ($a{=}10$). Cross-sectional slices of the 3D Helmholtz solution. Hermite-NGP accurately captures the oscillatory wave patterns, achieving an $L^2$ error of $6 \times 10^{-3}$, substantially better than the closest baseline, I-NGP-FD ($7.21 \times 10^{-2}$).

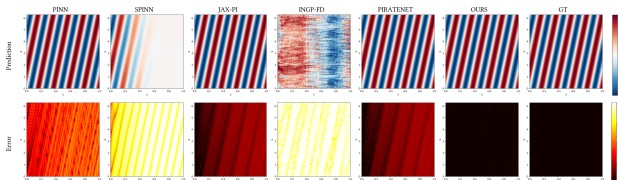

*Figure 5.* Convection 1+1D ($c = 30$): Solution field at final time. Hermite-NGP preserves the sharp traveling wave with $L^2 = 8.49 \times 10^{-5}$, a $10\times$ improvement over PirateNet ($8.54 \times 10^{-4}$), while SPINN and INGP-FD fail to converge.

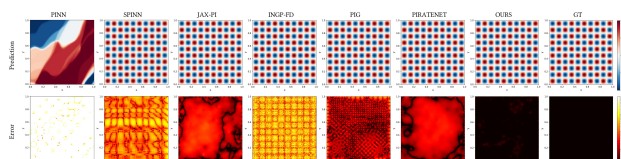

*Figure 6.* Helmholtz 2D ($a = 10$): Ground truth (left), Hermite-NGP prediction (middle), and pointwise error (right). Our method achieves relative $L^2$ error of $1.81 \times 10^{-5}$, compared to $3.57 \times 10^{-4}$ for the closest baseline

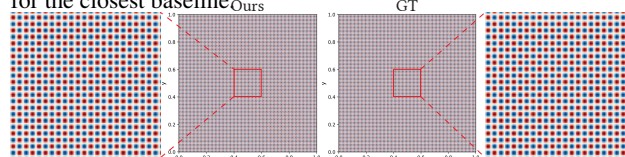

*Figure 7.* Helmholtz 2D ($a = 100$) enlarged: Hermite-NGP is the only method that converges on this challenging high-frequency setting, achieving $L^2 = 4.59 \times 10^{-2}$. All baseline methods fail to capture the rapid oscillations.

following Sitzmann et al. (2020) with $\omega_0 = 30$; hash table coefficients start near zero. Algorithm 1 summarizes the complete Hermite-NGP training pipeline, from hash encoding through analytic derivatives to PDE loss computation.

The recursion in (8)–(10) works with any twice-differentiable activation (Swish, Softplus, GELU, etc.); even with an autograd MLP, the Hermite encoding's analytic derivatives are preserved.

## 6. Multi-Resolution Coarse-to-Fine Training

Inspired by multigrid methods (Briggs et al., 2000) and coarse-to-fine strategies in neural surface reconstruction (Li et al., 2023b), we leverage the hierarchical structure of multi-resolution hash encoding for curriculum training. We employ a three-phase strategy: (1) train only coarse levels ($l = 0, \dots, L_0$) to capture global structure; (2) progressively activate finer levels; (3) fine-tune all levels jointly. The number of active levels follows $L_{\text{active}}(t) = \min(L, L_0 + \lfloor t/\tau \rfloor)$, where $\tau$ is the activation interval. Ablation studies (Section 7.7) show coarse-to-fine achieves 79% error reduction over training all levels simultaneously. We summarize the pipeline in Algorithm 1.

## 7. Experiments

We evaluate Hermite-NGP on a broad suite of 2D/3D PDE benchmarks, covering elliptic, hyperbolic, and parabolic equations, as well as geometric applications. These benchmarks probe complementary aspects of neural PDE solvers, including high-frequency oscillations, sharp transport, cou-

pled dynamics, complex geometries, and geometric constraints. For problems without analytic solutions, we obtain ground truth from high-resolution conventional PDE solvers. All metrics are reported as relative $L_2$ errors. Full PDE formulations and training details are provided in App. B.

### 7.1. Experimental Setup

**Baselines.** We compare Hermite-NGP against recent PINN-based methods: PirateNet (Wang et al., 2024b), JAX-PI (Wang et al., 2023a), PIG (Kang et al., 2024), INGP-FD (Huang & Alkhalifah, 2024) (hash encoding with finite differences), SPINN (Cho et al., 2023), PIXEL (Kang et al., 2023), and a vanilla PINN baseline (Raissi et al., 2019). For all baselines, we use the official implementations with recommended hyperparameters and, unless otherwise noted, a shared training protocol (Adam (Kingma, 2014) optimizer, GradNorm loss balancing), see details in Appendix B.2.

Table 1 summarizes the relative $L_2$ errors across all benchmarks, and full PDE formulations are provided in App. B.1.

### 7.2. 2D PDE Experiments

We evaluate two 2D benchmarks that stress high-frequency oscillations and sharp transport. For **Helmholtz 2D**

*Table 1.* Relative $L^2$ error comparison. Best results in **bold**. "fail" denotes failure ($L^2 \geq 1.0$ or divergence). NA indicates not evaluated. [†]Reported in the original paper. [‡]Compact format (larger model). For Helmholtz 2D, our results report small-model performance, with large-model results in parentheses.

| Method | Helmholtz 2D $a=10$ | Helmholtz 2D $a=20$ | Helmholtz 2D $a=100$ | Helmholtz 3D $a=3$ | Helmholtz 3D $a=10$ | Conv. $c=30$ | T-G $\nu=.01$ | Flow Mix. |
|---|---|---|---|---|---|---|---|---|
| Ours[‡] | 5.29e-05 (**1.81e-05**) | 9.87e-05 (**7.93e-05**) | **4.59e-02** | **6.09e-05** | **6.01e-03** | **8.49e-05** | **7.71e-05** | **2.35e-04** |
| PirateNet | 3.57e-04 | 1.36e-03 | fail | 8.40e-04 | 1.55e-01 | 8.54e-04 | fail | NA |
| JAX-PI | 5.76e-04 | 1.20e-03 | fail | NA | NA | 8.54e-04 | NA | NA |
| INGP-FD | 1.67e-03 | 2.77e-03 | 4.98e-01 | 4.04e-03 | 7.21e-02 | 7.02e-01 | 6.80e-01 | fail |
| SPINN | 5.46e-03 | 3.30e-02 | 7.08e-01 | 2.28e-02 | 9.61e-02 | 7.92e-01 | 3.98e-01 | 2.90e-03[†] |
| PIXEL | 3.47e-02 | 1.34e-01 | fail | NA | NA | 1.84e-03[†] | NA | NA |
| PINN | fail | fail | fail | 2.58e-01 | fail | 1.11e-02 | 4.57e-02 | NA |
| PIG | 7.04e-04 | fail | fail | 2.57e-01 | fail | NA | 7.27e-04 | 2.67e-04[†] |

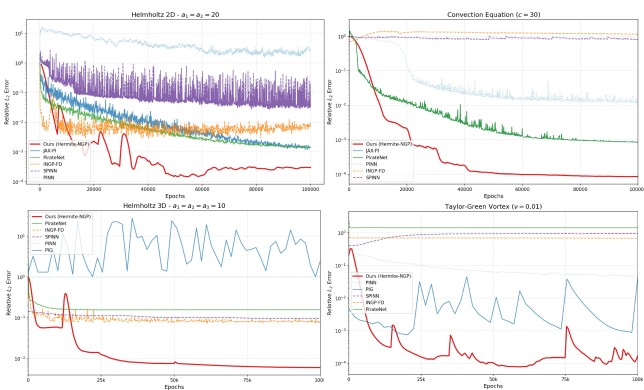

*Figure 8.* Training loss curves across four different benchmarks. Our method's loss curve decreases smoothly, in contrast to the highly oscillatory behavior of the baselines, and converges to errors up to **four orders of magnitude** lower across different benchmarks; see Table 1 for details.

(Figs. 6, 18, and 19), Hermite-NGP achieves relative $L^2$ errors of $1.81 \times 10^{-5}$, $7.93 \times 10^{-5}$, and $4.59 \times 10^{-2}$ for $a=10, 20, 100$, corresponding to $20\times$, $17\times$, and $11\times$ improvements over the strongest baselines, respectively. Hermite-NGP is also the only method that converges at the challenging $a=100$ setting. For **time-dependent Convection** (1+1**D**) with $c=30$ (Fig. 5), Hermite-NGP attains a relative error of $8.49 \times 10^{-5}$, a $10\times$ reduction compared to PirateNet ($8.54 \times 10^{-4}$). Grid-based methods (INGP-FD, SPINN) fail to resolve the sharp front due to severe numerical diffusion from their finite-difference discretization.

**Additional Baselines (Helmholtz 2D, $a=10$).** Under a unified protocol, $\partial^\infty$-Grid (Kairanda et al., 2026) reaches $6.07 \times 10^{-3}$ and SIREN (Sitzmann et al., 2020) $6.67 \times 10^{-2}$ – both two to three orders of magnitude behind our result in Table 1 (Fig. 9; Appendix Table 18).

### 7.3. 3D PDE Experiments

We evaluate three 3D benchmarks, including a stationary Helmholtz problem and two time-dependent 2+1D flows. For **Helmholtz 3D** (Figs. 4, 17), Hermite-NGP achieves relative $L^2$ errors of $6.09 \times 10^{-5}$ at $a=3$ ($14\times$

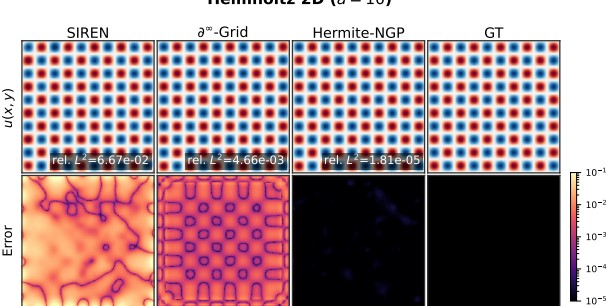

*Figure 9.* Helmholtz 2D ($a=10$) qualitative comparison across Hermite-NGP, $\partial^\infty$-Grid (Kairanda et al., 2026), SIREN. Only Hermite-NGP recovers the high-frequency structure cleanly.

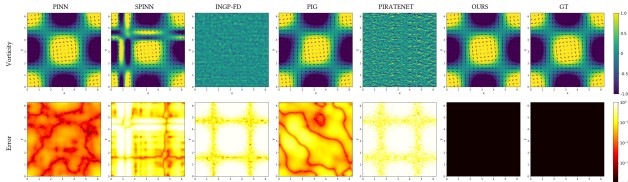

*Figure 10.* Taylor-Green vortex ($\nu = 0.01$): Velocity magnitude at $t = 1$. Hermite-NGP captures the decaying vortex structure with $L^2 = 7.71 \times 10^{-5}$, outperforming PIG by $9\times$.

lower than PirateNet) and $6.01 \times 10^{-3}$ at $a=10$ ($12\times$ lower than INGP-FD), demonstrating strong scalability to 3D. For **time-dependent Taylor–Green vortex** (2+1**D**) (Fig. 10), Hermite-NGP attains a relative $L^2$ error of $7.71 \times 10^{-5}$, which is $9\times$ lower than PIG and over $600\times$ lower than the vanilla PINN baseline, while PirateNet fails to converge on this coupled system. For **time-dependent flow mixing** (2+1**D**) (Figs. 11 and 16), Hermite-NGP reaches a relative $L^2$ error of $2.35 \times 10^{-4}$, slightly outperforming

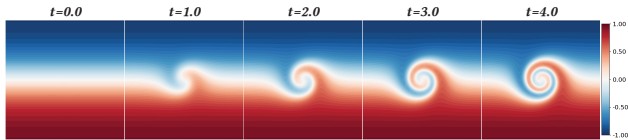

*Figure 11.* Flow Mixing: Solution field showing rotational transport. Hermite-NGP achieves $L^2 = 2.35 \times 10^{-4}$, resolving sharp gradients from fluid stretching.

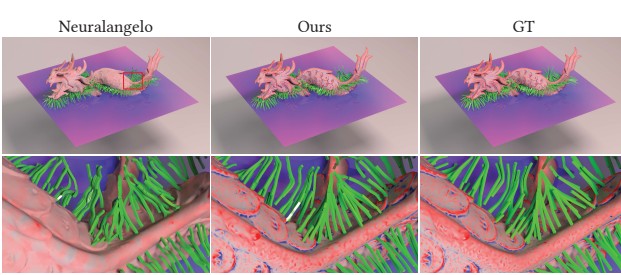

*Figure 12.* **SDF Learning and Curvature Estimation.** Hermite-NGP learns an SDF whose gradients (green line segments) and curvature (shown as color) remain smooth thanks to analytic second derivatives, whereas NeuralAngelo's finite-difference curvature is noisy and exhibits visible artifacts.

PIG ($2.67 \times 10^{-4}$) and achieving a $12\times$ improvement over SPINN; the PIG and SPINN numbers are taken from their original publications. For the first two benchmarks, the second-row error visualizations show that Hermite-NGP consistently has the smallest discrepancy from the ground truth compared to all baselines. Quantitative results are summarized in Table 1.

### 7.4. 3D PDEs on Complex Geometries

We evaluate Hermite-NGP on 3D Stanford meshes with varying geometric complexity, assessing the ability of neural PDE solvers to handle intricate shapes, complex boundaries, and higher-order geometric operators such as curvature.

**3D Poisson with Mesh Boundary Conditions.** We solve a homogeneous Poisson problem (Laplace equation) $\Delta u = 0$ on $[0,1]^3$, with the mesh surface as an inner Dirichlet boundary ($u = 1$) and the domain boundary as an outer Dirichlet boundary ($u = 0$). Ground truth is computed using a high-resolution conjugate-gradient solver in SciPy (Virtanen et al., 2020). Table 2 reports relative $L^2$ errors, where ours achieves, on average, a $3\times$ lower error than PIG.

**SDF learning.** Table 2 reports the mean absolute error (MAE) of the gradient, where Hermite-NGP attains a $2.4\times$ lower error than NeuralAngelo (Li et al., 2023b) on average. As shown in Fig. 12, our analytic second derivatives produce smooth and accurate curvature shown as color on mesh, whereas NeuralAngelo's finite-difference approximations yield noisy curvature with visible artifacts, a critical limitation for downstream tasks such as mesh extraction and physics simulation.

### 7.5. Image Reconstruction from Gradient Supervision

To show that the gradient-augmented hash representation generalizes beyond PDE residuals, we reconstruct the `camera` image from *gradient*-only supervision following Kairanda et al. (2026). Hermite-NGP attains the best PSNR at both resolutions (32.56 dB at $256\times256$, 32.35 dB at $512\times512$), ahead of $\partial^\infty$-Grid (Kairanda et al., 2026) and SIREN (Sitzmann et al., 2020) (Fig. 13; best result in Appendix Table 19).

*Table 2.* **3D with Complex Geometry.** Poisson with mesh boundary (MAE ↓) and SDF learning (Grad MAE ↓).

| Mesh | 3D Poisson: L2 ↓ | | SDF: Grad. MAE ↓ | |
|---|---|---|---|---|
| | Ours | PIG | Ours | NeuralAngelo |
| Armadillo | **0.0055** | 0.0167 | **0.0478** | 0.1009 |
| Bunny | **0.0044** | 0.0127 | **0.0416** | 0.0887 |
| Fandisk | **0.0031** | 0.0100 | **0.0516** | 0.1064 |
| Lucy | – | – | **0.0418** | 0.1213 |
| Dragon | – | – | **0.0453** | 0.1322 |
| Average | **0.0043** | 0.0131 | **0.0456** | 0.1099 |

**Supervision with $L_{grad}$**

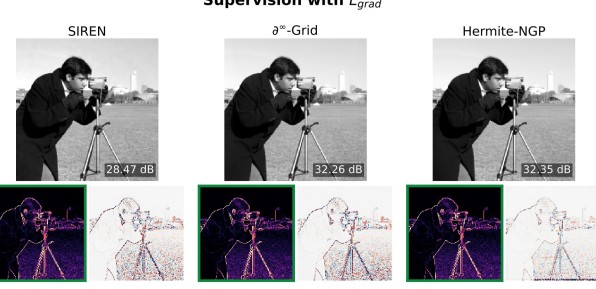

*Figure 13.* Image reconstruction from *gradient* supervision on the `camera` image. Hermite-NGP recovers sharper edges and finer texture than $\partial^\infty$-Grid and SIREN while matching their soft-region quality.

### 7.6. GPU Memory and Speed Comparison

Hermite-NGP scales efficiently: 68K to 16.8M parameters uses 133–389 MB memory and 1.8–3.6 ms/epoch (Table 10). INGP-FD with similar parameter counts (81K–11.2M) uses 5.4–241 MB and 2.5–8.7 ms/epoch. PIG scales poorly: 1600 Gaussians (32K params) requires 33.5 GB memory and 5 s/epoch. Full details are provided in Appendix Table 10.

**3D Training Memory vs. INGP-FD.** Despite the $2^d$ storage overhead, in 3D Hermite-NGP uses *less* peak GPU memory than INGP-FD across the tested range (150K–33.6M params), because INGP-FD's 7 forward passes for the central-difference Laplacian retain $7\times$ activation graphs while Hermite-NGP keeps a single graph (Appendix Table 17).

### 7.7. Ablation Studies

We conduct ablation studies on Helmholtz 2D ($a = 5, 10, 15, 20$) to analyze key design choices. We report key findings here and refer to Appendix C for complete results.

**Cubic-NGP Baseline.** We evaluate higher-order NGP variants (Trilinear, Cubic, Bicubic Catmull–Rom) that differentiate through the encoding without stored derivatives. On Helmholtz 2D ($a=10$), all variants fail (L2 > 0.1), including true $4\times4$ Bicubic, while Hermite-NGP achieves $1.81 \times 10^{-5}$. The underlying failure mode is that hash collisions inject high-frequency noise into the stored feature values; computing derivatives through (or by FD on) these

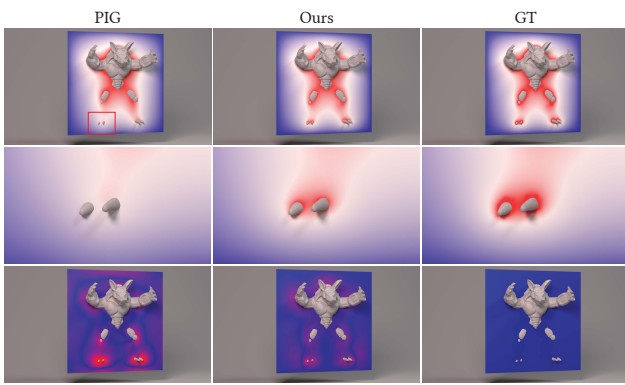

Figure 14. 3D Poisson with mesh boundary (Armadillo). Solution $u$ with Dirichlet values $u=1$ on the mesh surface and $u=0$ on the outer domain boundary. Hermite-NGP achieves an $L^2$ error of $5 \times 10^{-3}$, a $3\times$ improvement over PIG. The third row shows L2 error.

noisy features amplifies the noise. Hermite-NGP instead treats the derivative field as a first-class optimization target, distributing the representation burden across function and derivative channels. Figure 15 provides a visual comparison and confirms that stored derivative coefficients are essential.

**Coarse-to-Fine Training.** We evaluate several multigrid-inspired resolution schedules. Our coarse-to-fine (C2F) strategy reduces error by 79.2% relative to training without any scheduling, and consistently outperforms both V- and W-cycle variants, which yield smaller improvements. Detailed results for all 28 configurations are provided in Appendix C.1.

**Hash Table Allocation.** We ablate hash capacity for Hermite coefficients: values ($H_1$), first ($H_2$), and mixed ($H_3$) derivatives. The optimal 14–14–10 ($\log_2$) achieves $2.26 \times 10^{-5}$ L2 error, a 56% reduction over uniform 12–12–12. Reducing $H_2$ to 10 degrades error to $9.98\times10^{-5}$, while reducing $H_1$ yields $8.90 \times 10^{-5}$, showing higher collision sensitivity for first derivatives. Appendix C.2 reports all 30 settings.

**Architecture Parameters.** We ablate hash size, resolution levels, and per-level scale on 2D Helmholtz ($a=5, 10, 20$). Performance is frequency-dependent: at $a=5$, hash $2^{16}$ achieves $6.52 \times 10^{-6}$; at $a=10$, a compact model yields $5.29 \times 10^{-5}$, improved to $1.81 \times 10^{-5}$ with hash $2^{14}$; at $a=20$, a compact model yields $9.87 \times 10^{-5}$, improved to $7.93\times10^{-5}$ with hash $2^{16}$. Scale 2.0 and 8 levels perform robustly across frequencies. Full sweeps are in Appendix C.3.

**MLP Depth.** On Helmholtz 2D ($a=20$), depth 4 improves error by $1.7\times$ over the default depth 2 at a cost of $\sim0.9$ ms/epoch per layer; the hash encoding carries most of the representational burden, so we keep $d=2$ as default (Appendix Table 16).

**Computational Efficiency.** Training takes 2.85–3.13 ms per epoch with 10K collocation points, broken down into 0.35–0.41 ms for PDE residuals, 0.23–0.26 ms for boundary conditions, and 1.98–2.17 ms for the backward pass.

With 100K points, each epoch takes 19.7 ms, of which the backward pass accounts for 16.5 ms (84%). See Appendix C.4 for a full scaling analysis across 5K–100K collocation points.

**Robustness Across Seeds.** Across 5 seeds on every main benchmark, relative standard deviations stay below $\sim15\%$ (e.g., Helmholtz 2D $a=10$: $3.35\times10^{-5}\pm4.4\times10^{-6}$; $a=100$: $7.34\times10^{-2} \pm 6.06\times10^{-3}$), confirming the gains are not seed-driven (Appendix Table 15).

**Derivative Computation.** We compare full analytic derivatives to autograd-based alternatives. Our method achieves 3–15$\times$ speedup (9.5$\times$ avg.) over full autograd. A hybrid with analytic encoding but autograd MLP is only 1.2–1.5$\times$ slower, indicating analytic encoding derivatives dominate the speedup. Gains are largest at small batches (15$\times$ at 5K collocation) due to near-constant autograd overhead. Full results are in Appendix C.4.

## 8. Conclusion

We presented **Hermite-NGP**, a gradient-augmented multi-resolution hash encoding for neural PDE solving. By explicitly storing function values and mixed partial derivatives at hash grid vertices and reconstructing the field via Hermite interpolation, Hermite-NGP enables analytic, stable evaluation of first- and second-order differential operators, overcoming the discontinuities of standard NGP encodings. The resulting $C^1$-continuous representation, combined with spatial adaptivity and coarse-to-fine training, yields an efficient neural PDE solver for problems on complex geometries.

**Limitations and Future Work.** Hermite-NGP incurs storage overhead from explicit mixed derivative parameterization, which may grow in higher dimensions, though offset by reduced training memory from analytic derivatives. Our implementation uses SIREN for efficient second-order derivatives; extending to other activation families remains future work. We also plan to explore quintic Hermite interpolation for PDEs requiring higher-order derivatives. Our Hermite interpolation is currently realized over multi-resolution hash encodings under strong-form PDE residuals; weak-form variants are left as future work, and our method is not intended as a substitute for classical FDM/FEM solvers.

## Acknowledgments

We sincerely thank the anonymous reviewers for their valuable feedback. Georgia Tech authors acknowledge NSF CAREER #2420319, IIS #2433307, OISE #2433313, IIS #2433322, ECCS #2318814, and CNS #2450401 for funding support, and the NVIDIA Academic Grant Program for hardware support. We credit the Houdini education license for rendering.

## Impact Statement

The goal of this work is to improve the reliability and efficiency of neural PDE solvers by providing analytic and faithful spatial derivatives, particularly in settings involving complex geometries and higher-order differential operators. We do not foresee direct negative societal consequences arising from this work.

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

## A. Analytic Derivative Computation

Here, we provide complete derivations of the analytic derivative formulas used in Section 5.1 of the main text. Our neural graphics primitives (NGP) network architecture consists of two components: a Hermite interpolation–based multi-resolution representation and a SIREN-based MLP network. We first present the analytic derivatives for the Hermite interpolation–based multi-resolution representation, and then derive the corresponding formulas for the SIREN network. Finally, we combine these results to obtain the complete analytic derivatives of the full model.

### A.1. Full Derivation of Encoding Derivatives

We derive the first and second derivatives of the Hermite encoding in detail, culminating in a complete 2D worked example.

**Setup.** Consider a single resolution level with grid spacing $\Delta x$. For a query point $\mathbf{x} = (x, y)$ in 2D, let the enclosing cell have corners at $(x_0, y_0)$, $(x_0 + \Delta x, y_0)$, $(x_0, y_0 + \Delta x)$, $(x_0 + \Delta x, y_0 + \Delta x)$. We denote corners as: $(0, 0) \rightarrow$ bottom-left, $(1, 0) \rightarrow$ bottom-right, $(0, 1) \rightarrow$ top-left, $(1, 1) \rightarrow$ top-right and define local coordinates:

$$s = \frac{x - x_0}{\Delta x}, \quad t = \frac{y - y_0}{\Delta x}, \quad s, t \in [0, 1]. \tag{11}$$

**2D Hermite Interpolant.** The bicubic Hermite interpolant is:

$$H[f](s, t) = \sum_{i,j \in \{0,1\}} \Big[ f_{ij}^{(00)} H^{(00)}(s - i, t - i) + f_{ij}^{(10)} H^{(10)}(s - i, t - i) \Delta x \\ + f_{ij}^{(01)} H^{(01)}(s - i, t - i) \Delta x + f_{ij}^{(11)} H^{(11)}(s - i, t - i) \Delta^2 x \Big] \tag{12}$$

where $f_{ij}$, $f_{ij}^{(10)}$, $f_{ij}^{(01)}$, $f_{ij}^{(11)}$ are the function value, $x$-derivative, $y$-derivative, and mixed derivative at corner $(i, j)$, and the basis functions are tensor products:

$$H^{(00)}(s, t) = h^{(0)}(s) h^{(0)}(t), \tag{13}$$

$$H^{(10)}(s, t) = h^{(1)}(s) h^{(0)}(t), \tag{14}$$

$$H^{(01)}(s, t) = h^{(0)}(s) h^{(1)}(t), \tag{15}$$

$$H^{(11)}(s, t) = h^{(1)}(s) h^{(1)}(t). \tag{16}$$

**First Derivatives.** Using the chain rule $\partial/\partial x = (1/\Delta x)\partial/\partial s$:

$$\frac{\partial H[f]}{\partial x} = \frac{1}{\Delta x} \sum_{i,j \in \{0,1\}} \Big[ f_{ij}^{(00)} h^{(0)\prime}(s - i) h^{(0)}(t - i) + f_{ij}^{(10)} h^{(1)\prime}(s - i) h^{(0)}(t - i) \Delta x \\ + f_{ij}^{(01)} h^{(0)\prime}(s - i) h^{(1)}(t - i) \Delta x + f_{ij}^{(11)} h^{(1)\prime}(s - i) h^{(1)}(t - i) \Delta^2 x \Big] \tag{17}$$

Simplifying the $\Delta x$ factors:

$$\frac{\partial H[f]}{\partial x} = \sum_{i,j \in \{0,1\}} \Big[ \frac{1}{\Delta x} f_{ij}^{(00)} h^{(0)\prime}(s - i) h^{(0)}(t - i) + f_{ij}^{(10)} h^{(1)\prime}(s - i) h^{(0)}(t - i) \\ + f_{ij}^{(01)} h^{(0)\prime}(s - i) h^{(1)}(t - i) + f_{ij}^{(11)} h^{(1)\prime}(s - i) h^{(1)}(t - i) \Delta x \Big] \tag{18}$$

**Second Derivatives.** Taking another derivative:

$$\frac{\partial^2 H[f]}{\partial x^2} = \sum_{i,j \in \{0,1\}} \Big[ \frac{1}{\Delta^2 x} f_{ij}^{(00)} h^{(0)\prime}(s - i) h^{(0)}(t - i) + \frac{1}{\Delta x} f_{ij}^{(10)} h^{(1)\prime}(s - i) h^{(0)}(t - i) \\ + \frac{1}{\Delta x} f_{ij}^{(01)} h^{(0)\prime}(s - i) h^{(1)}(t - i) + f_{ij}^{(11)} h^{(1)\prime}(s - i) h^{(1)}(t - i) \Big] \tag{19}$$

**Complete 2D Example with All 16 Terms.** Here, we expand Equation (12). The full interpolant has 16 terms (4 corners $\times$ 4 coefficients each):

$$H[f](s,t) =$$

$$f_{ij}^{(00)}h^{(0)}(s)h^{(0)}(t) + f_{ij}^{(00)}h^{(0)}(s-1)h^{(0)}(t) + f_{ij}^{(00)}h^{(0)}(s)h^{(0)}(t-1) + f_{ij}^{(00)}h^{(0)}(s-1)h^{(0)}(t-1)+$$

$$f_{ij}^{(10)}h^{(1)}(s)h^{(0)}(t)\Delta x + f_{ij}^{(10)}h^{(1)}(s-1)h^{(0)}(t)\Delta x + f_{ij}^{(10)}h^{(1)}(s)h^{(0)}(t-1)\Delta x + f_{ij}^{(10)}h^{(1)}(s-1)h^{(0)}(t-1)\Delta x+$$

$$f_{ij}^{(01)}h^{(0)}(s)h^{(1)}(t)\Delta x + f_{ij}^{(01)}h^{(0)}(s-1)h^{(1)}(t)\Delta x + f_{ij}^{(01)}h^{(0)}(s)h^{(1)}(t-1)\Delta x + f_{ij}^{(01)}h^{(0)}(s-1)h^{(1)}(t-1)\Delta x+$$

$$f_{ij}^{(11)}h^{(1)}(s)h^{(1)}(t)\Delta^2 x + f_{ij}^{(11)}h^{(1)}(s-1)h^{(1)}(t)\Delta^2 x+$$

$$f_{ij}^{(11)}h^{(1)}(s)h^{(1)}(t-1)\Delta^2 x + f_{ij}^{(11)}h^{(1)}(s-1)h^{(1)}(t-1)\Delta^2 x+$$

$$(20)$$

This formula makes explicit how each of the $2^2 \times 2^2 = 16$ coefficients stored in the hash tables contributes to the interpolated value.

**Laplacian Computation.** The Laplacian $\nabla^2 H[f] = \partial^2 H[f]/\partial x^2 + \partial^2 H[f]/\partial y^2$ is computed by evaluating (19) for both directions and summing. Each term involves:

- Retrieving 16 coefficients from hash tables (4 corners $\times$ 4 derivative types)

- Evaluating 4 second-derivative basis values: $h^{(0)\prime\prime}(s), h^{(0)\prime\prime}(s-1), h^{(1)\prime\prime}(s), h^{(1)\prime\prime}(s-1)$ (and similarly for $t$)

- Computing tensor products and weighted sums

The factored structure allows efficient implementation: compute 1D basis values once per dimension, then form outer products.

### A.2. Multi-Layer SIREN Laplacian

We prove by induction that the Laplacian propagates efficiently through a $K$-layer SIREN network, reusing forward pass quantities.

**Proposition A.1** (SIREN Derivative Propagation). *Let $u(\mathbf{x}) = W_{K+1}a^{(K)} + b_{K+1}$ be a $K$-hidden-layer SIREN network with $a^{(0)} = \gamma(\mathbf{x})$ (the Hermite encoding) and $a^{(k)} = \sin(\omega z^{(k)})$, $z^{(k)} = W_k a^{(k-1)} + b_k$ for $k = 1, \ldots, K$. Then the spatial derivatives satisfy:*

$$\frac{\partial a^{(k)}}{\partial x_i} = \omega \cos(\omega z^{(k)}) \odot W_k \frac{\partial a^{(k-1)}}{\partial x_i}, \tag{21}$$

$$\frac{\partial^2 a^{(k)}}{\partial x_i^2} = -\omega^2 a^{(k)} \odot \left( W_k \frac{\partial a^{(k-1)}}{\partial x_i} \right)^2 + \omega \cos(\omega z^{(k)}) \odot W_k \frac{\partial^2 a^{(k-1)}}{\partial x_i^2}, \tag{22}$$

*where $\odot$ denotes element-wise multiplication.*

*Proof.* We proceed by induction on layer index $k$.

**Base case** ($k = 0$)**:** The encoding derivatives $\partial\gamma/\partial x_i$ and $\partial^2\gamma/\partial x_i^2$ are computed analytically from the Hermite basis as derived in Appendix A.1.

**Inductive step:** Assume (21)–(22) hold for layer $k - 1$. For layer $k$:

*First derivative:* By the chain rule,

$$\frac{\partial a^{(k)}}{\partial x_i} = \frac{\partial}{\partial x_i} \sin(\omega z^{(k)})$$

$$= \omega \cos(\omega z^{(k)}) \odot \frac{\partial z^{(k)}}{\partial x_i}$$

$$= \omega \cos(\omega z^{(k)}) \odot W_k \frac{\partial a^{(k-1)}}{\partial x_i},$$

using $\partial z^{(k)}/\partial x_i = W_k \partial a^{(k-1)}/\partial x_i$ (since $b_k$ is constant).

*Second derivative:* Differentiating again,

$$
\begin{aligned}
\frac{\partial^2 a^{(k)}}{\partial x_i^2} &= \frac{\partial}{\partial x_i}\left[\omega \cos(\omega z^{(k)}) \odot W_k \frac{\partial a^{(k-1)}}{\partial x_i}\right] \\
&= -\omega^2 \sin(\omega z^{(k)}) \odot \frac{\partial z^{(k)}}{\partial x_i} \odot W_k \frac{\partial a^{(k-1)}}{\partial x_i} + \omega \cos(\omega z^{(k)}) \odot W_k \frac{\partial^2 a^{(k-1)}}{\partial x_i^2} \\
&= -\omega^2 a^{(k)} \odot \left(W_k \frac{\partial a^{(k-1)}}{\partial x_i}\right)^2 + \omega \cos(\omega z^{(k)}) \odot W_k \frac{\partial^2 a^{(k-1)}}{\partial x_i^2}.
\end{aligned}
$$

This completes the induction. The final output derivative is:

$$
\frac{\partial^2 u}{\partial x_i^2} = W_{K+1} \frac{\partial^2 a^{(K)}}{\partial x_i^2},
$$

and the Laplacian is $\nabla^2 u = \sum_{i=1}^{d} \partial^2 u/\partial x_i^2$. $\qquad\square$

*Remark* A.2 (Computational Reuse). The key efficiency insight is that all quantities in (21)–(22) are available from a modified forward pass:

- $a^{(k)} = \sin(\omega z^{(k)})$ is the standard activation

- $\cos(\omega z^{(k)})$ can be computed alongside $\sin(\omega z^{(k)})$ at negligible extra cost

- $W_k \partial a^{(k-1)}/\partial x_i$ uses the same weight matrices as the forward pass

No backward pass or automatic differentiation graph is needed.

### A.3. Mixed Partial Derivatives

For PDEs involving mixed derivatives (e.g., Burgers equation, convection-diffusion), we derive the full Hessian matrix.

**Mixed Encoding Derivative.** The mixed second derivative of the Hermite encoding is:

$$
\frac{\partial^2 H[f]}{\partial^2 x \partial y} = \frac{1}{\Delta^2 x} \sum_{i,j}\left[f_{ij}^{(00)} h^{(0)\prime}(s-i) h^{(0)\prime}(t-j) + \cdots + f_{ij}^{(11)} \Delta x \Delta y h^{(1)\prime}(s-i) h^{(1)\prime}(t-j)\right], \tag{23}
$$

where we use first derivatives of basis functions in both directions.

**Mixed Derivative Through SIREN.** For the full network, the mixed derivative propagates as:

$$
\frac{\partial^2 a^{(k)}}{\partial x_i \partial x_j} = -\omega^2 a^{(k)} \odot \left(W_k \frac{\partial a^{(k-1)}}{\partial x_i}\right) \odot \left(W_k \frac{\partial a^{(k-1)}}{\partial x_j}\right) + \omega \cos(\omega z^{(k)}) \odot W_k \frac{\partial^2 a^{(k-1)}}{\partial x_i \partial x_j}. \tag{24}
$$

This allows computing the full Hessian matrix $\nabla^2 u$ analytically, which is needed for:

- Burgers equation: $u_t + u \cdot \nabla u = \nu \nabla^2 u$

- Navier-Stokes: requires both Laplacian and convective derivatives

- Geometric operators: Gaussian curvature involves determinant of Hessian

## B. Experimental Details

This appendix provides complete details for reproducing our experimental results.

## B.1. PDE Formulations

### B.1.1. HELMHOLTZ 2D EQUATION

The 2D Helmholtz equation models time-harmonic wave propagation:

$$-\Delta u(x,y) + k^2 u(x,y) = f(x,y), \quad (x,y) \in \Omega = [0,1]^2, \tag{25}$$

with Dirichlet boundary conditions $u = g$ on $\partial\Omega$. The wave number $k = 1$ is fixed; the frequency parameter $a_1$ controls solution oscillation. We use the manufactured solution:

$$u^*(x,y) = \sin(a_1 \pi x) \sin(a_2 \pi y), \tag{26}$$

with $a_2 = a_1$. Substituting into (25) yields the source term:

$$f(x,y) = \left(k^2 - (a_1^2 + a_2^2)\pi^2\right) \sin(a_1 \pi x) \sin(a_2 \pi y). \tag{27}$$

Higher $a_1 \in \{10, 20, 100\}$ produces increasingly oscillatory solutions that challenge neural network spectral bias.

### B.1.2. HELMHOLTZ 3D EQUATION

The 3D extension tests scalability to higher dimensions:

$$-\Delta u(x,y,z) + k^2 u(x,y,z) = f(x,y,z), \quad (x,y,z) \in \Omega = [0,1]^3, \tag{28}$$

with manufactured solution:

$$u^*(x,y,z) = \sin(a\pi x) \sin(a\pi y) \sin(a\pi z), \tag{29}$$

and corresponding source:

$$f(x,y,z) = \left(k^2 - 3a^2\pi^2\right) \sin(a\pi x) \sin(a\pi y) \sin(a\pi z). \tag{30}$$

We test $a \in \{3, 10\}$. The $2^d = 8$ Hermite coefficients per vertex in 3D represent a larger storage overhead, making this a stress test for our approach.

### B.1.3. CONVECTION 1+1D EQUATION

The 1D convection (advection) equation describes transport phenomena:

$$\frac{\partial u}{\partial t} + c\frac{\partial u}{\partial x} = 0, \quad x \in [0, 2\pi], \ t \in [0,1], \tag{31}$$

with periodic boundary conditions $u(0,t) = u(2\pi, t)$ and initial condition $u(x,0) = \sin(x)$. The exact solution is:

$$u^*(x,t) = \sin(x - ct). \tag{32}$$

High velocity $c = 30$ creates sharp propagating features that require accurate first-derivative computation. For neural network training, coordinates are normalized to $[0,1]^2$.

### B.1.4. TAYLOR-GREEN VORTEX 2+1D (NAVIER-STOKES)

The incompressible Navier-Stokes equations in 2D with time:

$$\frac{\partial u}{\partial t} + u\frac{\partial u}{\partial x} + v\frac{\partial u}{\partial y} = -\frac{\partial p}{\partial x} + \nu\left(\frac{\partial^2 u}{\partial x^2} + \frac{\partial^2 u}{\partial y^2}\right), \tag{33}$$

$$\frac{\partial v}{\partial t} + u\frac{\partial v}{\partial x} + v\frac{\partial v}{\partial y} = -\frac{\partial p}{\partial y} + \nu\left(\frac{\partial^2 v}{\partial x^2} + \frac{\partial^2 v}{\partial y^2}\right), \tag{34}$$

$$\frac{\partial u}{\partial x} + \frac{\partial v}{\partial y} = 0 \quad \text{(incompressibility)}. \tag{35}$$

Domain: $t \in [0, 1]$, $(x, y) \in [0, 2\pi]^2$. The Taylor-Green vortex provides an analytic solution:

$$u^*(x, y, t) = -\cos(x)\sin(y)\exp(-2\nu t), \tag{36}$$

$$v^*(x, y, t) = \sin(x)\cos(y)\exp(-2\nu t), \tag{37}$$

$$p^*(x, y, t) = -\tfrac{1}{4}\big(\cos(2x) + \cos(2y)\big)\exp(-4\nu t). \tag{38}$$

This coupled system requires accurate computation of both first derivatives (advection, pressure gradient, incompressibility) and second derivatives (viscous diffusion).

### B.1.5. FLOW MIXING 2+1D

A nonlinear advection equation with spatially-varying rotational velocity:

$$\frac{\partial u}{\partial t} + a(x, y)\frac{\partial u}{\partial x} + b(x, y)\frac{\partial u}{\partial y} = 0, \tag{39}$$

where the velocity field describes mixing flow with angular velocity depending on radial distance:

$$r = \sqrt{x^2 + y^2}, \quad v_t = \frac{\tanh(r)}{\cosh^2(r)}, \tag{40}$$

$$\omega = \frac{v_t/v_{\max}}{r}, \quad a = -\frac{v_t}{v_{\max}} \cdot \frac{y}{r}, \quad b = \frac{v_t}{v_{\max}} \cdot \frac{x}{r}. \tag{41}$$

Domain: $t \in [0, 4]$, $(x, y) \in [-4, 4]^2$. The exact solution:

$$u^*(t, x, y) = -\tanh\left(\frac{y}{2}\cos(\omega t) - \frac{x}{2}\sin(\omega t)\right). \tag{42}$$

This problem develops sharp gradients as fluid parcels stretch and fold under the mixing dynamics.

For both Poisson 3D and SDF learning experiments, we evaluate on five Stanford meshes of varying geometric complexity:

- **Armadillo**: High-genus mesh with intricate surface details

- **Bunny**: Classic benchmark with moderate complexity

- **Fandisk**: CAD model with sharp edges and flat regions

- **Lucy**: High-resolution scan with fine geometric features

- **Dragon (xyzrgb)**: Complex topology with thin structures

### B.1.6. POISSON 3D WITH MESH BOUNDARY

We solve the Laplace equation in a domain with an embedded mesh surface:

$$\Delta u = 0, \quad \mathbf{x} \in \Omega \setminus \mathcal{M}, \tag{43}$$

where $\Omega = [0, 1]^3$ and $\mathcal{M}$ is one of the five Stanford meshes. Boundary conditions:

$$u = 1 \quad \text{on } \partial\mathcal{M} \text{ (mesh surface)}, \tag{44}$$

$$u = 0 \quad \text{on } \partial\Omega \text{ (domain boundary)}. \tag{45}$$

The solution is a harmonic potential field smoothly varying from 1 on the mesh surface to 0 on the domain boundary. Ground truth is computed using a high-resolution finite element solver. This benchmark tests handling of complex geometry without body-fitted meshes.

### B.1.7. SDF LEARNING

We learn a signed distance function $u$ in $\Omega = [0,1]^3$ around a mesh surface $\partial\mathcal{M}$ using direct supervision against ground-truth SDF values and gradients sampled from a high-resolution reference. The loss combines a value term and a gradient term:

$$\mathcal{L}_{\text{sdf}} = \|u - u_{\text{gt}}\|^2, \tag{46}$$

$$\mathcal{L}_{\text{grad}} = \|\nabla u - \nabla u_{\text{gt}}\|^2. \tag{47}$$

This tests first-derivative accuracy in a purely geometric setting where the gradient field is the primary quantity of interest.

### B.2. Training Configuration

Table 3 summarizes the collocation point configurations for each benchmark.

*Table 3.* Collocation point configuration for each PDE benchmark.

| PDE | Setting | Interior Points | BC/IC Points | Epochs |
|-----|---------|-----------------|--------------|--------|
| *Elliptic PDEs* | | | | |
| Helmholtz 2D | $a_1 = 10$ | 10,000 | 5,000/edge | 100k |
| Helmholtz 2D | $a_1 = 20$ | 40,000 | 8,000/edge | 100k |
| Helmholtz 2D | $a_1 = 100$ | 100,000 | 25,000/edge | 100k |
| Helmholtz 3D | $a = 3$ | 40,000 | 20,000 | 100k |
| Helmholtz 3D | $a = 10$ | 40,000 | 20,000 | 100k |
| *Time-Dependent PDEs* | | | | |
| Convection 1+1D | $c = 30$ | 10,000 | 5,000 (IC), 5,000 (BC) | 100k |
| Taylor-Green 2+1D | $\nu = 0.01$ | 20,000 | 5,000 (IC) | 100k |
| Flow Mixing 2+1D | – | 50,000 | 10,000 (IC), 3,000/edge | 200k |
| *Complex Geometry (5 meshes)* | | | | |
| Poisson 3D | mesh $\rightarrow 1$, boundary $\rightarrow 0$ | 50,000 | 20,000 (5,000/face) | – |
| SDF learning | – | 50,000 | – | – |

**Initialization.** Following Sitzmann et al. (2020), MLP weights are initialized from $\mathcal{U}(-\sqrt{6/n}, \sqrt{6/n})$ with $\omega_0 = 30$ for the first layer. Hermite hash table function values are initialized with $\mathcal{N}(0, 0.01)$; derivative coefficients are initialized to zero, representing locally constant functions.

**Optimizer and Learning Rate.** All experiments use the Adam optimizer with initial learning rate $10^{-3}$ and cosine annealing schedule with warm restarts. The restart period is 10,000 iterations with restart multiplier 2 (i.e., periods of 10K, 20K, 40K, ...). Collocation points and total epochs for each benchmark are summarized in Table 3.

**Loss Balancing.** We use an adaptive gradient-based loss balancing scheme inspired by GradNorm (Chen et al., 2018). At each iteration, we compute the gradient norms of the PDE residual loss ($g_{\text{pde}}$) and boundary condition loss ($g_{\text{bc}}$). When $g_{\text{bc}} > 10^{-8}$, the BC weight is updated via exponential moving average:

$$\lambda_{\text{bc}} \leftarrow 0.9 \cdot \lambda_{\text{bc}} + 0.1 \cdot \frac{g_{\text{pde}}}{g_{\text{bc}}}, \quad \lambda_{\text{bc}} \in [1, \lambda_{\max}], \tag{48}$$

where $\lambda_{\max}$ is a problem-dependent cap. This balances gradient magnitudes between PDE and BC losses during training.

**Exponential Moving Average.** We maintain an EMA of model parameters with decay rate 0.999 for stable evaluation. Final reported errors use the EMA model weights.

**Coarse-to-Fine Training.** For the multi-resolution curriculum (Section 4.4), we use the Coarse-to-Fine (Equal) strategy: all $L$ resolution levels are divided into equal-duration phases, with levels activated progressively from coarse to fine. Each level is trained for $\tau = T/L$ iterations before the next level is activated, where $T$ is the total training iterations.

**Hash Table Configuration.** For Helmholtz 2D, we select configurations based on our ablation study (Section 7.7). For other 2D benchmarks and 3D problems with simple geometry, we use default settings: hash table size $T = 2^{14}$, $L = 8$ resolution levels, and level ratio $r = 1.5$. For 3D problems with complex geometry (Poisson with mesh boundary, SDF learning), we increase capacity to $T = 2^{16}$, $L = 8$ levels, and level ratio $r = 2.0$ to better capture fine geometric details.

**MLP Architecture.** For baseline methods, we use MLPs with hidden dimension 128 and 4 layers, except SPINN which uses its default configuration. For Hermite-NGP, we use hidden dimension 128 with 2 layers, as our hash encoding stores additional derivative coefficients; the shallower MLP balances overall memory usage while the encoding provides sufficient representational capacity. In speed benchmarks (Table 10), all methods use identical MLP configurations (128 hidden, 2 layers) for fair comparison.

**Hardware.** All experiments are conducted on a single NVIDIA 4090 GPU with 24GB memory.

### B.3. Baseline Implementation Details

All baselines use official implementations with recommended hyperparameters from the original papers:

- **PirateNet** (Wang et al., 2024b): Official JAX implementation with 4 layers, 128 neurons.

- **JAX-PI** (Wang et al., 2023a): Official implementation with RAR (Residual-based Adaptive Refinement) sampling, 4 layers, 128 neurons.

- **PIG** (Kang et al., 2024): 3000 Gaussian particles with learnable positions and covariances (maximum GPU capacity).

- **INGP-FD** (Huang & Alkhalifah, 2024): Same hash encoding hyperparameters as Hermite-NGP, but with standard bilinear interpolation and finite-difference Laplacian ($\epsilon = 10^{-3}$), 4 layers, 128 neurons.

- **SPINN** (Cho et al., 2023): Separable basis with default configuration (100 functions per axis, 8 layers).

- **PIXEL** (Kang et al., 2023): $256 \times 256$ learnable grid for 2D problems, 4 layers, 128 neurons.

- **PINN** (Raissi et al., 2019): Standard MLP with 4 layers, 128 neurons, tanh activations.

- **NeuralAngelo** (Li et al., 2023b): Multi-resolution hash encoding with finite difference for SDF learning, 2 layers, 128 neurons. We use $T = 2^{18}$ for NeuralAngelo and $T = 2^{16}$ for Hermite-NGP to match parameter counts in SDF/curvature experiments 7.3.

- $\partial^{\infty}$**-Grid** (Kairanda et al., 2026): Official implementation with RBF interpolation on co-located feature grids. For Helmholtz 2D ($a$=10) we use grid resolution 256 with 4 scales, learning rate $10^{-3}$, and hard boundary conditions; for image reconstruction we use the authors' default `grid_rbf_gradient_image.ini` configuration at the matching resolution.

- **SIREN** (Sitzmann et al., 2020): 4-layer, 128-neuron MLP with sinusoidal activations and $\omega_0 = 30$ initialization. Used as a coordinate-network baseline for the Helmholtz $a$=10 and image-reconstruction experiments.

All baselines are trained with the same optimizer settings (Adam, cosine annealing) and loss balancing (GradNorm) as Hermite-NGP to ensure fair comparison.

## C. Ablation Studies

We conduct comprehensive ablation studies on Helmholtz 2D to analyze the impact of key design choices.

### C.1. Coarse-to-Fine Training Ablation

We compare 28 curriculum learning configurations across different strategy types and phase durations. Table 4 shows results sorted by best L2 error.

**Analysis.** Coarse-to-fine training significantly outperforms all other strategies, achieving 79% error reduction over baseline (1.20e-4 vs 5.76e-4). V-cycle and W-cycle strategies, while better than baseline, suffer from "forgetting" when returning to coarse levels. Fine-to-coarse training performs poorly as high-frequency noise in early stages corrupts learning. Warmup, gradual ramping, and oscillating strategies underperform baseline, suggesting that abrupt level transitions are beneficial.

*Table 4.* Curriculum learning ablation on Helmholtz 2D ($a = 15$). All experiments use 100K total epochs.

| Curriculum Type | Phase Ep | Best $L_2$ | Final $L_2$ | Time (s) |
|---|---|---|---|---|
| Coarse-to-Fine | 20K | $1.20 \times 10^{-4}$ | $1.05 \times 10^{-4}$ | 261.0 |
| Coarse-to-Fine | 40K | $1.97 \times 10^{-4}$ | $1.36 \times 10^{-4}$ | 264.1 |
| Coarse-to-Fine | 20K | $2.03 \times 10^{-4}$ | $1.40 \times 10^{-4}$ | 269.0 |
| Coarse-to-Fine | 20K | $2.22 \times 10^{-4}$ | $1.58 \times 10^{-4}$ | 266.3 |
| Coarse-to-Fine | 30K | $2.44 \times 10^{-4}$ | $1.73 \times 10^{-4}$ | 264.8 |
| Coarse-to-Fine | 20K | $2.51 \times 10^{-4}$ | $1.80 \times 10^{-4}$ | 268.4 |
| V-Cycle | 20K | $2.75 \times 10^{-4}$ | $1.94 \times 10^{-4}$ | 270.6 |
| V-Cycle | 20K | $2.75 \times 10^{-4}$ | $1.92 \times 10^{-4}$ | 265.2 |
| W-Cycle | 20K | $2.80 \times 10^{-4}$ | $1.96 \times 10^{-4}$ | 265.3 |
| V-Cycle | 10K | $2.89 \times 10^{-4}$ | $2.01 \times 10^{-4}$ | 272.2 |
| W-Cycle | 20K | $3.04 \times 10^{-4}$ | $2.09 \times 10^{-4}$ | 267.5 |
| Coarse-to-Fine | 15K | $3.14 \times 10^{-4}$ | $2.16 \times 10^{-4}$ | 266.3 |
| W-Cycle | 15K | $3.18 \times 10^{-4}$ | $2.24 \times 10^{-4}$ | 267.1 |
| Coarse-to-Fine | 10K | $3.31 \times 10^{-4}$ | $2.27 \times 10^{-4}$ | 265.5 |
| Sandwich | 20K | $3.32 \times 10^{-4}$ | $2.48 \times 10^{-4}$ | 268.9 |
| V-Cycle | 25K | $3.34 \times 10^{-4}$ | $2.57 \times 10^{-4}$ | 269.4 |
| V-Cycle | 15K | $3.72 \times 10^{-4}$ | $2.60 \times 10^{-4}$ | 269.9 |
| V-Cycle | 5K | $4.52 \times 10^{-4}$ | $3.31 \times 10^{-4}$ | 269.6 |
| W-Cycle | 25K | $4.69 \times 10^{-4}$ | $3.23 \times 10^{-4}$ | 267.1 |
| Coarse-to-Fine | 5K | $4.95 \times 10^{-4}$ | $3.40 \times 10^{-4}$ | 263.8 |
| None (Baseline) | 20K | $5.76 \times 10^{-4}$ | $4.01 \times 10^{-4}$ | 264.5 |
| Fine-to-Coarse | 20K | $6.07 \times 10^{-4}$ | $4.15 \times 10^{-4}$ | 264.6 |
| W-Cycle | 10K | $6.26 \times 10^{-4}$ | $4.33 \times 10^{-4}$ | 269.6 |
| Warmup + C2F | 20K | $6.30 \times 10^{-4}$ | $4.33 \times 10^{-4}$ | 263.6 |
| C2F Gradual | 20K | $7.14 \times 10^{-4}$ | $5.69 \times 10^{-4}$ | 265.1 |
| Oscillate | 20K | $9.31 \times 10^{-4}$ | $6.49 \times 10^{-4}$ | 267.1 |

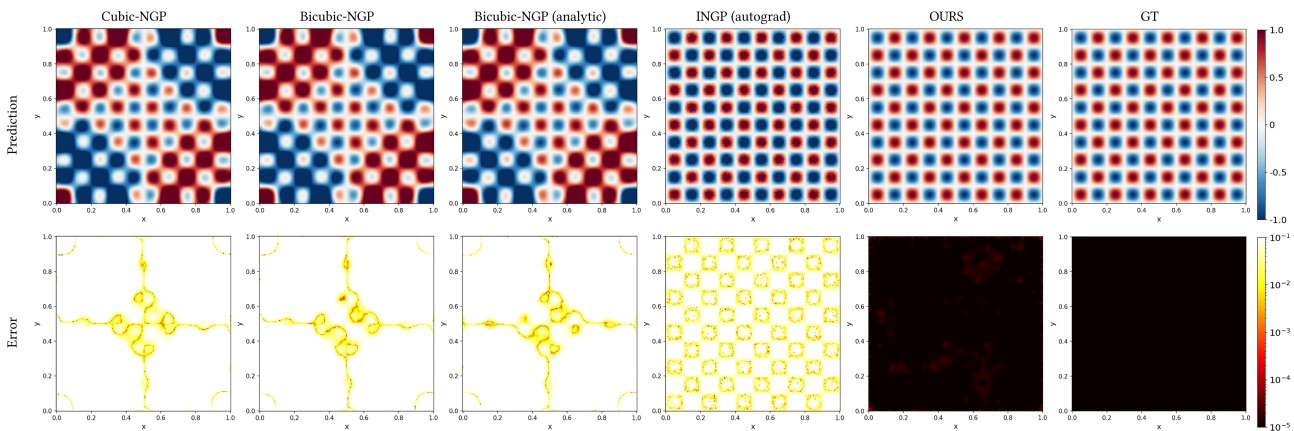

*Figure 15.* **Approximate Ablation.** Comparison of encoding variants on Helmholtz ($a = 10$). Trilinear, Cubic, and Bicubic NGP all fail (L2 > 0.1), while Hermite-NGP achieves 1.81e-5, confirming stored derivative coefficients are essential.

*Table 5.* Complete hash table allocation ablation. $H_1$, $H_2$, $H_3$ are $\log_2$ of hash table sizes for $u$, $\partial u/\partial x$, $\partial u/\partial y$ respectively.

| $H_1$ | $H_2$ | $H_3$ | Params | Best $L_2$ | Rank | $H_1$ | $H_2$ | $H_3$ | Params | Best $L_2$ | Rank |
|---|---|---|---|---|---|---|---|---|---|---|---|
| 14 | 14 | 10 | 822K | 2.26e-5 | 1 | 10 | 12 | 12 | 232K | 5.41e-5 | 16 |
| 14 | 12 | 12 | 478K | 2.40e-5 | 2 | 14 | 10 | 14 | 576K | 5.73e-5 | 17 |
| 16 | 12 | 12 | 1.26M | 2.52e-5 | 3 | 8 | 14 | 14 | 809K | 7.61e-5 | 18 |
| 10 | 16 | 16 | 3.18M | 2.64e-5 | 4 | 12 | 12 | 8 | 220K | 8.27e-5 | 19 |
| 12 | 14 | 12 | 674K | 2.93e-5 | 5 | 10 | 12 | 14 | 428K | 8.51e-5 | 20 |
| 14 | 12 | 10 | 428K | 3.03e-5 | 6 | 16 | 12 | 8 | 1.20M | 8.78e-5 | 21 |
| 12 | 16 | 10 | 2.20M | 3.47e-5 | 7 | 10 | 14 | 10 | 576K | 8.90e-5 | 22 |
| 10 | 14 | 14 | 822K | 3.63e-5 | 8 | 14 | 10 | 10 | 330K | 9.98e-5 | 23 |
| 8 | 16 | 16 | 3.17M | 3.66e-5 | 9 | 8 | 12 | 16 | 1.20M | 1.08e-4 | 24 |
| 12 | 12 | 16 | 1.26M | 4.00e-5 | 10 | 12 | 10 | 12 | 183K | 1.13e-4 | 25 |
| 12 | 12 | 10 | 232K | 4.62e-5 | 11 | 16 | 8 | 8 | 1.08M | 1.21e-4 | 26 |
| 12 | 16 | 12 | 2.25M | 4.82e-5 | 12 | 8 | 12 | 12 | 220K | 1.22e-4 | 27 |
| 12 | 12 | 14 | 478K | 5.11e-5 | 13 | 12 | 10 | 16 | 1.17M | 1.57e-4 | 28 |
| 12 | 12 | 12 | 281K | 5.17e-5 | 14 | 10 | 10 | 14 | 330K | 1.66e-4 | 29 |
| 16 | 10 | 10 | 1.12M | 5.18e-5 | 15 | 12 | 8 | 12 | 158K | 1.75e-4 | 30 |

## C.2. Hash Table Allocation Ablation

We study how to allocate hash table capacity across the three Hermite coefficient types: values ($H_1$), first derivatives ($H_2$: $\partial u/\partial x$), and second derivatives ($H_3$: $\partial u/\partial y$). Table 5 shows all 30 configurations tested on Helmholtz 2D ($a = 15$), ranked by L2 error.

**Analysis.** Several key patterns emerge from the complete results:

- $H_2$ **(first derivatives) is most critical**: The top configuration uses $H_2$=14. Reducing $H_2$ from 14 to 10 increases error from 2.26e-5 to 9.98e-5.

- $H_3$ **can be smaller than $H_1$, $H_2$**: The optimal $H_3$=10 outperforms larger $H_3$ values, suggesting the $\partial u/\partial y$ component has lower spatial variation.

- $H_1$ **(values) has moderate sensitivity**: Reducing $H_1$ from 14 to 10 causes 294% degradation (8.90e-5 vs 2.26e-5).

- **Non-uniform allocation is optimal**: The best $(14, 14, 10)$ uses only 822K parameters, outperforming the larger uniform $(16, 16, 16)$ configuration.

*Table 6.* Architecture ablation on Helmholtz 2D ($a = 5$). Top 10 configurations by L2 error.

| Hash | Scale | Lvls | Params | Best $L_2$ | Final $L_2$ |
|------|-------|------|--------|-----------|------------|
| 16 | 2.0 | 8 | 4.21M | 6.52e-6 | 6.46e-6 |
| 14 | 2.0 | 8 | 1.07M | 8.34e-6 | 8.23e-6 |
| 12 | 2.0 | 8 | 281K | 8.70e-6 | 8.52e-6 |
| 10 | 1.5 | 10 | 101K | 9.65e-6 | 8.63e-6 |
| 14 | 1.5 | 8 | 1.07M | 1.23e-5 | 1.19e-5 |
| 14 | 1.3 | 8 | 1.07M | 1.72e-5 | 1.63e-5 |
| 14 | 1.8 | 8 | 1.07M | 1.73e-5 | 1.68e-5 |
| 10 | 1.5 | 8 | 84K | 1.95e-5 | 1.78e-5 |
| 14 | 2.0 | 6 | 805K | 1.98e-5 | 1.95e-5 |
| 10 | 2.0 | 8 | 84K | 2.05e-5 | 1.86e-5 |

*Table 7.* Architecture ablation on Helmholtz 2D ($a = 10$). Top 10 configurations by L2 error.

| Hash | Scale | Lvls | Params | Best $L_2$ | Final $L_2$ |
|------|-------|------|--------|-----------|------------|
| 14 | 2.0 | 8 | 1.07M | 1.81e-5 | 1.66e-5 |
| 16 | 2.0 | 8 | 4.21M | 3.84e-5 | 3.83e-5 |
| 12 | 2.0 | 8 | 281K | 3.90e-5 | 3.37e-5 |
| 14 | 2.0 | 10 | 1.33M | 4.63e-5 | 3.81e-5 |
| 14 | 2.0 | 6 | 805K | 5.29e-5 | 5.08e-5 |
| 14 | 1.8 | 8 | 1.07M | 5.93e-5 | 5.39e-5 |
| 14 | 1.5 | 8 | 1.07M | 6.40e-5 | 5.95e-5 |
| 14 | 1.8 | 6 | 805K | 9.72e-5 | 8.46e-5 |
| 10 | 1.5 | 8 | 84K | 1.07e-4 | 9.10e-5 |
| 10 | 1.5 | 10 | 101K | 1.33e-4 | 1.08e-4 |

## C.3. Architecture Parameter Ablation

We ablate hash table size, number of resolution levels, and per-level scale factor across Helmholtz 2D at different frequencies ($a = 5, 10, 20$). This reveals how optimal architecture depends on PDE frequency content.

### C.3.1. HELMHOLTZ 2D ($a = 20$) – HIGH FREQUENCY

**Analysis.**

- **Hash table size**: Higher frequencies ($a = 20$) benefit more from larger hash tables ($2^{16}$), while medium frequencies ($a = 10$) work well with $2^{14}$. This suggests high-frequency solutions have finer spatial structure requiring more hash capacity.

*Table 8.* Architecture ablation on Helmholtz 2D ($a = 20$). Top 10 configurations by L2 error.

| Hash | Scale | Lvls | Params | Best $L_2$ | Final $L_2$ |
|------|-------|------|--------|-----------|------------|
| 16 | 2.0 | 8 | 4.21M | 7.93e-5 | 6.97e-5 |
| 14 | 1.5 | 8 | 1.07M | 3.33e-4 | 2.65e-4 |
| 14 | 2.0 | 6 | 805K | 3.37e-4 | 2.89e-4 |
| 14 | 2.3 | 8 | 1.07M | 5.00e-4 | 4.40e-4 |
| 12 | 2.0 | 8 | 281K | 5.53e-4 | 4.60e-4 |
| 14 | 1.5 | 6 | 805K | 5.68e-4 | 4.57e-4 |
| 14 | 2.0 | 8 | 1.07M | 6.38e-4 | 6.32e-4 |
| 14 | 2.0 | 10 | 1.33M | 6.56e-4 | 5.42e-4 |
| 14 | 1.8 | 8 | 1.07M | 6.64e-4 | 5.47e-4 |
| 14 | 1.8 | 6 | 805K | 1.37e-3 | 1.21e-3 |

*Table 9.* Optimal configuration comparison across frequencies.

| $a$ | Hash | Scale | Lvls | Params | Best $L_2$ |
|---|---|---|---|---|---|
| 5 | 16 | 2.0 | 8 | 4.21M | 6.52e-6 |
| 10 | 14 | 2.0 | 8 | 1.07M | 1.81e-5 |
| 20 | 16 | 2.0 | 8 | 4.21M | 7.93e-5 |

*Table 10.* GPU memory comparison at 10K collocation points. Our method does not apply GradNorm, reducing backward-pass memory and computation. In contrast, I-NGP-FD relies on tiny-cuda-nn, which constrains the hash-to-level ratio and requires more levels when using larger hash tables, increasing memory usage.

| Method | Params | Peak GPU (MB) | ms/epoch |
|---|---|---|---|
| Hermite-NGP ($2^{10}$) | 68K | 133 | 1.82 |
| Hermite-NGP ($2^{12}$) | 264K | 133 | 1.99 |
| Hermite-NGP ($2^{14}$) | 1.05M | 154 | 1.86 |
| Hermite-NGP ($2^{16}$) | 4.20M | 195 | 2.01 |
| Hermite-NGP ($2^{18}$) | 16.8M | 389 | 3.62 |
| INGP-FD ($2^{12}$, L=8) | 81K | 5.4 | 2.52 |
| INGP-FD ($2^{14}$, L=8) | 130K | 6.3 | 2.57 |
| INGP-FD ($2^{16}$, L=10) | 492K | 14.0 | 5.03 |
| INGP-FD ($2^{18}$, L=12) | 2.33M | 52.5 | 6.37 |
| INGP-FD ($2^{20}$, L=14) | 11.2M | 241.1 | 8.66 |
| PIG (200 Gaussians) | 4K | 4,230 | 47.0 |
| PIG (400 Gaussians) | 8K | 8,390 | 96.1 |
| PIG (800 Gaussians) | 16K | 16,760 | 189.6 |
| PIG (1600 Gaussians) | 32K | 33,480 | 4994.3 |

- **Scale and levels**: Scale 2.0 with 8 levels consistently performs well across all frequencies. This combination provides good frequency coverage from coarse to fine scales.

- **Speed-accuracy tradeoff**: At $a = 10$, the optimal $2^{14}$ configuration achieves comparable accuracy to $2^{16}$ while being $4\times$ smaller and faster (3.08 vs 3.40 ms/ep).

### C.4. Computational Scaling

We benchmark training speed across varying numbers of collocation points and hash sizes, with all timings measured on a single NVIDIA RTX 4090 GPU. For Table 10, GradNorm is not applied, reducing backward-pass time; all other speed benchmarks include GradNorm.

**Effect of Hash Table Size on Speed.** Table 12 shows timing at 10K collocation points for different hash sizes.

**Analysis.** The backward pass dominates computation time, accounting for 68% at 5K points and increasing to 84% at 100K points. This is expected as gradient computation scales with batch size. The forward pass (PDE + BC) scales sub-linearly

*Table 11.* Computational scaling: time per epoch (ms) and component breakdown. Hash=16, 8 levels.

| Collocation | PDE (ms) | BC (ms) | Backward (ms) | Total | Backward % |
|---|---|---|---|---|---|
| 5K | 0.33 | 0.23 | 1.83 | 2.70 | 68% |
| 10K | 0.38 | 0.26 | 2.17 | 3.13 | 69% |
| 15K | 0.50 | 0.29 | 2.48 | 3.60 | 69% |
| 20K | 0.60 | 0.26 | 3.11 | 4.28 | 73% |
| 25K | 0.78 | 0.27 | 3.80 | 5.15 | 74% |
| 50K | 1.35 | 0.25 | 7.30 | 9.19 | 79% |
| 100K | 2.59 | 0.28 | 16.48 | 19.72 | 84% |

*Table 12.* Speed vs. hash table size at 20K collocation points.

| Hash | Params | PDE (ms) | BC (ms) | Backward (ms) | Total |
|---|---|---|---|---|---|
| 8 | 35K | 0.37 | 0.23 | 2.02 | 2.89 |
| 10 | 84K | 0.40 | 0.26 | 2.47 | 3.42 |
| 12 | 281K | 0.41 | 0.25 | 2.16 | 3.12 |
| 14 | 1.07M | 0.38 | 0.26 | 2.17 | 3.13 |
| 16 | 4.21M | 0.36 | 0.24 | 2.09 | 3.04 |
| 18 | 16.8M | 0.38 | 0.23 | 2.40 | 4.42 |

*Table 13.* Derivative computation method comparison (hash=14, levels=8, hidden=128, layers=2). Use GradNorm

| Collocation | Ours (ms) | Autograd-MLP (ms) | Fully-Autograd (ms) | Speedup |
|---|---|---|---|---|
| 5K | 2.70 | 3.78 | 40.83 | 15.1× |
| 10K | 3.13 | 4.68 | 42.07 | 13.4× |
| 20K | 4.28 | 5.15 | 44.01 | 10.3× |
| 50K | 9.19 | 10.70 | 48.77 | 5.3× |
| 100K | 19.72 | 23.16 | 64.50 | 3.3× |

due to efficient hash table lookups. Hash table size has minimal impact on speed for sizes $2^8$ to $2^{16}$, with timing varying only from 2.89 to 3.04 ms/epoch at 10K points. Only at $2^{18}$ (16.8M params) does memory bandwidth become a bottleneck.

**Derivative Computation Ablation.**   We compare three derivative computation strategies to validate our analytic derivative approach:

- **Ours (Full Analytic)**: CUDA Hermite encoding with analytic derivatives + CUDA MLP with analytic Laplacian

- **Autograd-MLP**: CUDA Hermite encoding with analytic derivatives + PyTorch autograd for MLP Laplacian

- **Fully-Autograd**: Standard NGP encoding + PyTorch autograd for all derivatives

Our full analytic implementation achieves **3–15× speedup** over fully-autograd baselines (average 9.5×). The Autograd-MLP hybrid is only 1.2–1.5× slower than our approach, demonstrating that **analytic encoding derivatives provide the majority of the speedup**. Speedup is highest at smaller batch sizes (15× at 5K) because autograd overhead is relatively constant while computation scales linearly.

*Table 14.* Interpolation ablation on Helmholtz 2D ($a = 10$). Cubic uses smoothstep, Bicubic uses Catmull-Rom. All NGP variants fail.

| Method | Best $L_2$ | ms/epoch |
|---|---|---|
| Hermite-NGP (ours) | **1.81e-5** | 3.08 |
| Trilinear-NGP | 1.46e-1 | – |
| Cubic-NGP | 2.78e-1 | 693.6 |
| Bicubic-NGP | 2.91e-1 | 683.3 |
| Bicubic-NGP (analytic) | 2.47e-1 | 470.1 |

# D. Additional Experiments

This appendix reports the supporting numbers for experiments summarized in the main text: multi-seed robustness (Table 15), MLP depth ablation (Table 16), 3D training-memory comparison vs. INGP-FD (Table 17), additional baselines on Helmholtz 2D ($a$=10) (Table 18), and image reconstruction (Table 19). For image reconstruction, SIREN and $\partial^\infty$-Grid use Kairanda's percentile-rescale at evaluation (their convention); Hermite-NGP uses raw clip-to-$[0, 1]$ since its Dirichlet anchor fixes the integration constant.

*Table 15.* Multi-seed (5 seeds) relative $L^2$ error mean $\pm$ std on the main benchmarks, using the default configuration (hash $2^{14}$, $L=8$, scale 2.0, width 128, depth 2). Relative standard deviations stay below $\sim$15% across all settings.

| Benchmark | Relative $L^2$ (mean $\pm$ std) |
|---|---|
| Helmholtz 2D ($a=10$) | $3.35\times10^{-5} \pm 4.4\times10^{-6}$ |
| Helmholtz 2D ($a=20$) | $1.07\times10^{-4} \pm 3.64\times10^{-5}$ |
| Helmholtz 2D ($a=100$) | $7.34\times10^{-2} \pm 6.06\times10^{-3}$ |
| Helmholtz 3D ($a=10$) | $7.29\times10^{-3} \pm 1.10\times10^{-3}$ |
| Taylor–Green ($\nu=0.01$) | $5.69\times10^{-5} \pm 3.7\times10^{-6}$ |
| Convection ($c=30$) | $9.47\times10^{-5} \pm 2.5\times10^{-5}$ |

*Table 16.* MLP depth ablation on Helmholtz 2D ($a=20$, width 128, hash $2^{14}$). $^{*}$ paper default.

| Depth | Best $L^2$ | ms/epoch |
|---|---|---|
| 1 | $1.02\times10^{-3}$ | 2.17 |
| 2$^{*}$ | $6.81\times10^{-4}$ | 3.07 |
| 3 | $4.27\times10^{-4}$ | 3.95 |
| 4 | $4.03\times10^{-4}$ | 4.86 |

# E. Additional Figures

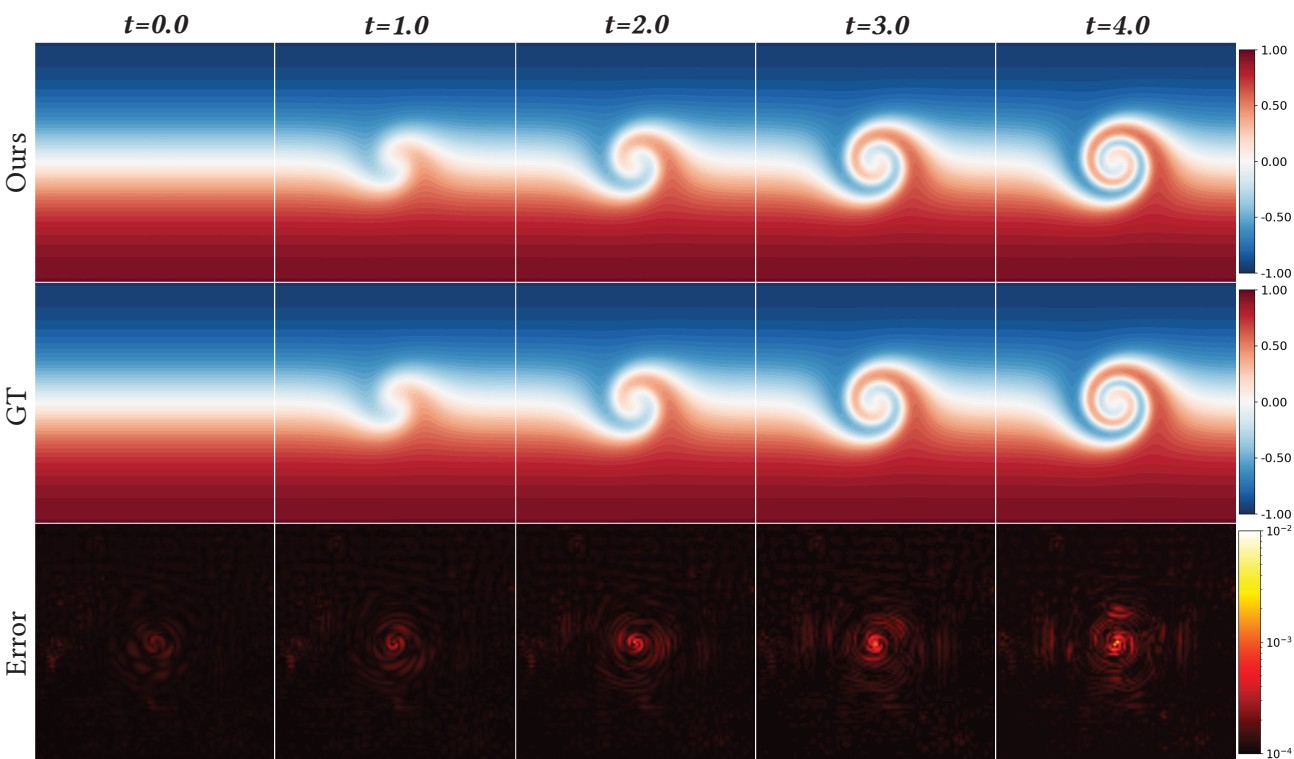

*Figure 16.* **Flow Mixing.** Solution snapshots at $t = 0, 0.2, 0.4, 0.6, 0.8, 1.0$. Hermite-NGP achieves average L2 error of 2.35e-4 across all time steps.

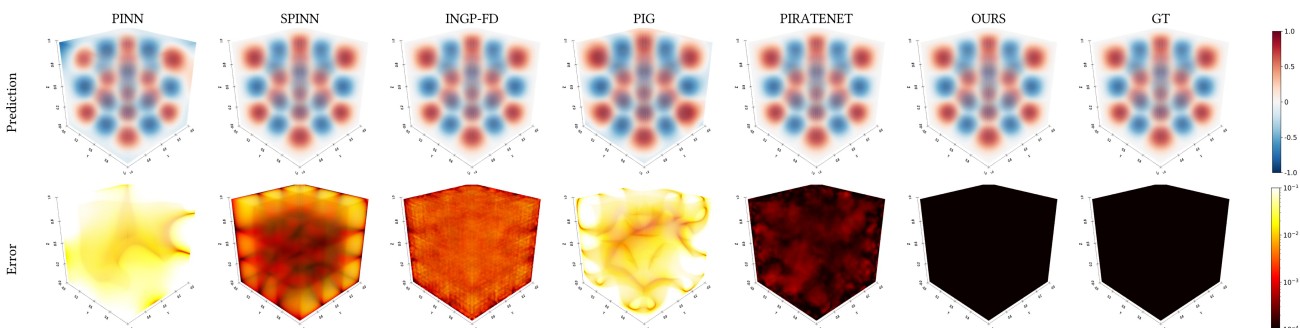

*Figure 17.* **Helmholtz 3D** ($a = 3$)**.** Cross-sectional slices of the 3D Helmholtz solution. Hermite-NGP achieves L2 error of 6.09e-5, compared to 8.40e-4 for PirateNet ($13.8\times$ lower).

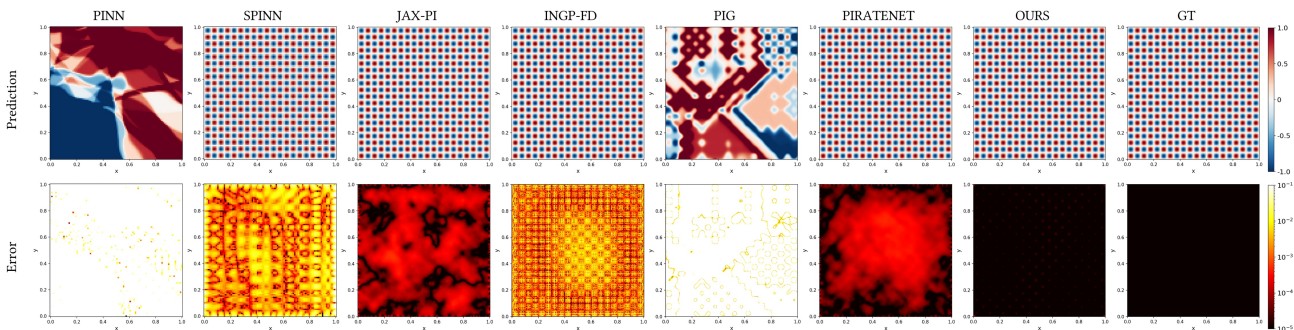

*Figure 18.* **Helmholtz 2D** ($a = 20$)**.** High-frequency solution with 20 wavelengths. Hermite-NGP achieves 9.87e-5 (1.07M params) and 7.93e-5 (4.21M params), outperforming PirateNet (1.36e-3), JAX-PI (1.2e-3), INGP-FD (2.77e-3), and SPINN (3.30e-2).

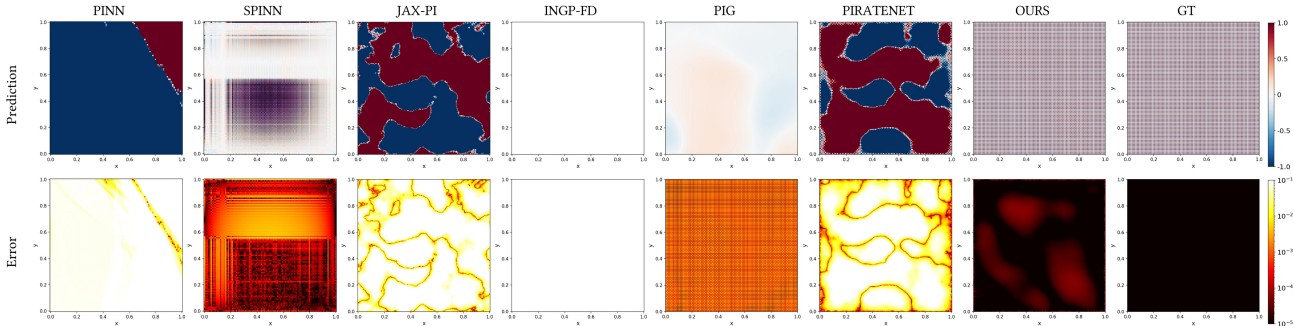

*Figure 19.* Helmholtz 2D ($a = 100$): Hermite-NGP is the only method that converges on this challenging high-frequency setting, achieving $L^2 = 4.59 \times 10^{-2}$. All baseline methods fail to capture the rapid oscillations.

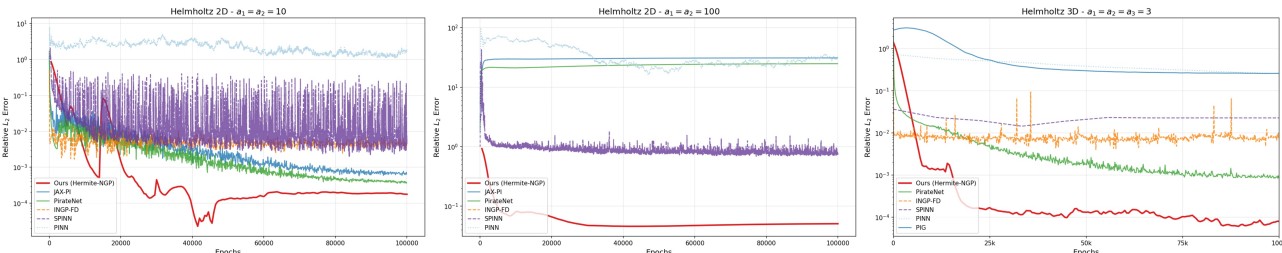

*Figure 20.* Training loss curves across three benchmarks. The horizontal axis indicates the number of epochs, while the vertical axis reports the relative $\ell_2$ training error. Our method exhibits a smoothly decreasing loss trajectory, in contrast to the strongly oscillatory behavior of the baselines, and converges to training errors up to **four orders of magnitude** smaller across the benchmarks; see Table 1 for details.

*Table 17.* Peak GPU training memory in 3D (Helmholtz 3D, $a=10$, 40k collocation points). Hermite-NGP uses less peak memory than INGP-FD across the full range because INGP-FD requires 7 forward passes (storing 7 activation graphs) while Hermite-NGP keeps a single graph.

| Hermite Params | Hermite Mem | INGP-FD Params | INGP-FD Mem |
|---:|---:|---:|---:|
| 150K | 957 MB | 117K | 1402 MB |
| 2.12M | 988 MB | 314K | 1405 MB |
| 8.41M | 1084 MB | 4.77M | 1502 MB |
| 33.6M | 1465 MB | 21.0M | 1783 MB |

*Table 18.* Additional baselines on Helmholtz 2D ($a=10$), all run under our unified protocol (Adam, cosine LR, identical collocation budget). Companion to Figure 9 in the main text.

| Method | Relative $L^2$ |
|---|---|
| Hermite-NGP (ours) | $3.35 \times 10^{-5}$ |
| INGP-FD (Huang & Alkhalifah, 2024) | $3.66 \times 10^{-3}$ |
| $\partial^{\infty}$-Grid (Kairanda et al., 2026) | $6.07 \times 10^{-3}$ |
| SIREN (Sitzmann et al., 2020) | $6.67 \times 10^{-2}$ |
| Cubic I-NGP | $2.47 \times 10^{-1}$ |

PIG        Ours        GT

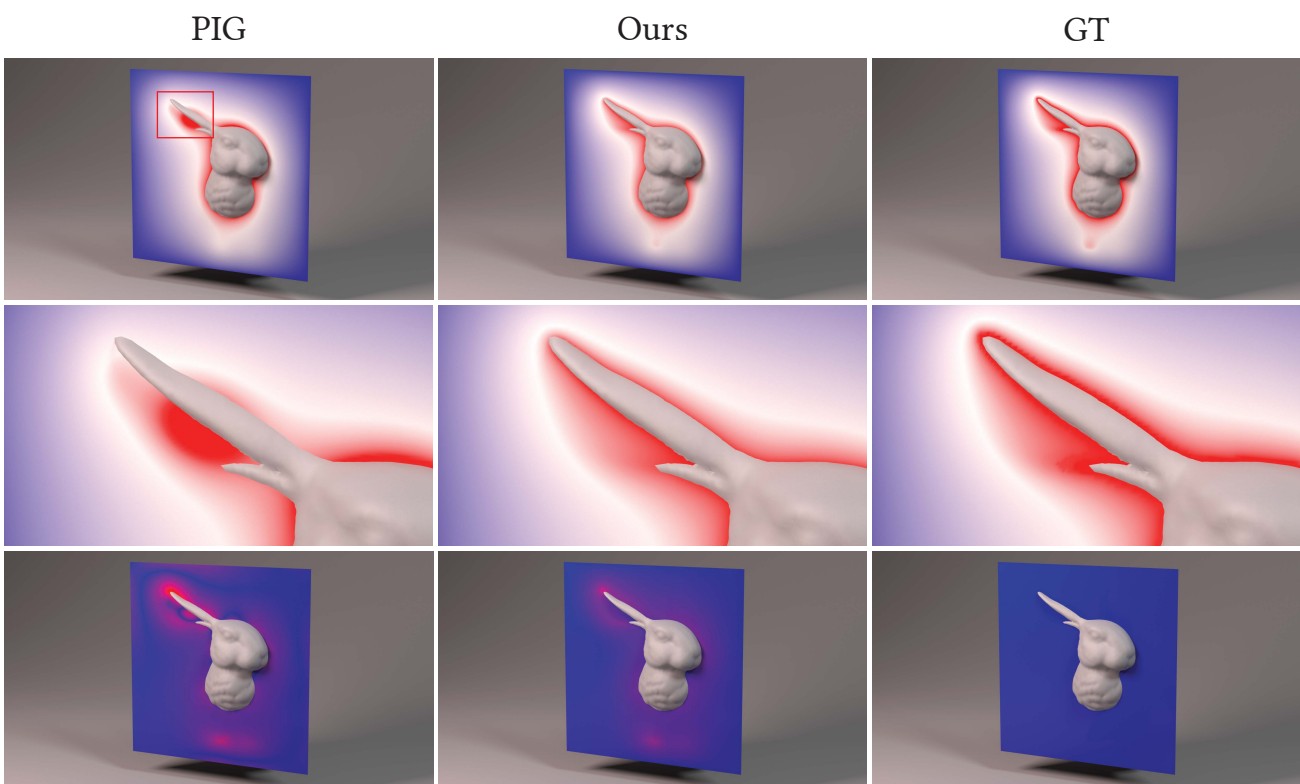

*Figure 21.* **Poisson 3D (Bunny).** Cross-sectional slice with mesh boundary and zoom-in detail. Hermite-NGP achieves MAE 0.0044 vs. PIG's 0.0127 (2.9× lower), with more accurate field near the boundary.

*Table 19.* Image reconstruction from *gradient* supervision on `camera`: best PSNR (dB) at two resolutions. Companion to Figure 13 in the main text.

| Method | 256×256 | 512×512 |
|---|---|---|
| Hermite-NGP (ours) | **32.56** | **32.35** |
| $\partial^\infty$-Grid (Kairanda et al., 2026) | 32.47 | 32.27 |
| SIREN (Sitzmann et al., 2020) | 32.47 | 29.25 |

| PIG | Ours | GT |
|---|---|---|

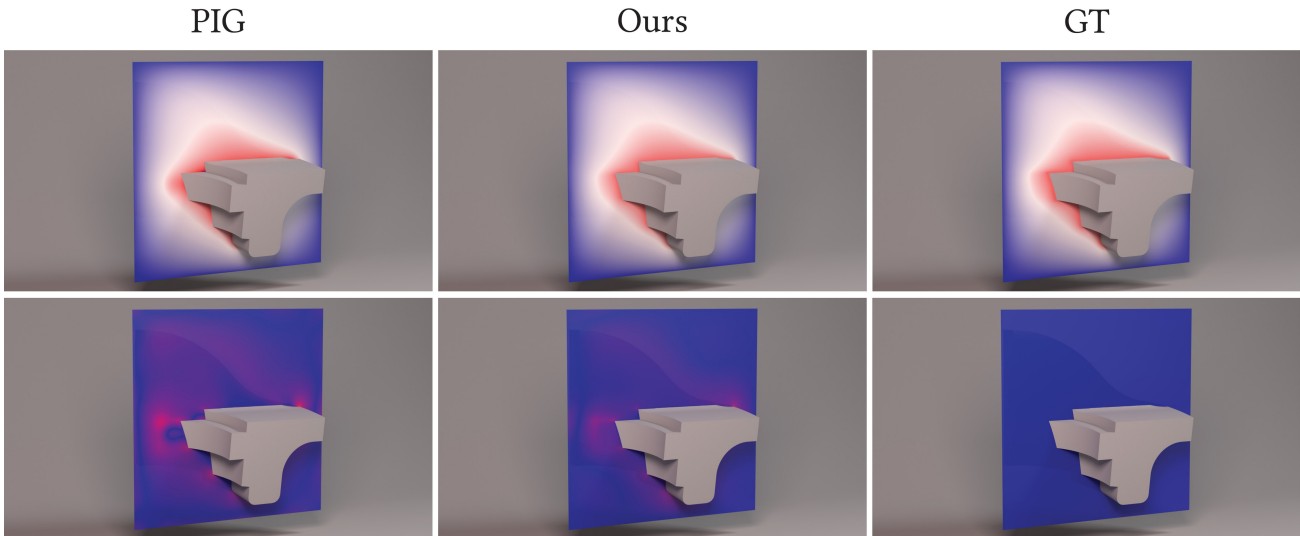

*Figure 22.* **Poisson 3D (Fandisk).** Cross-sectional slice with mesh boundary. Hermite-NGP achieves MAE 0.0031 vs. PIG's 0.0100 (3.2× lower), accurately capturing sharp geometric features.

| Neuralangelo | Ours | GT |
|---|---|---|

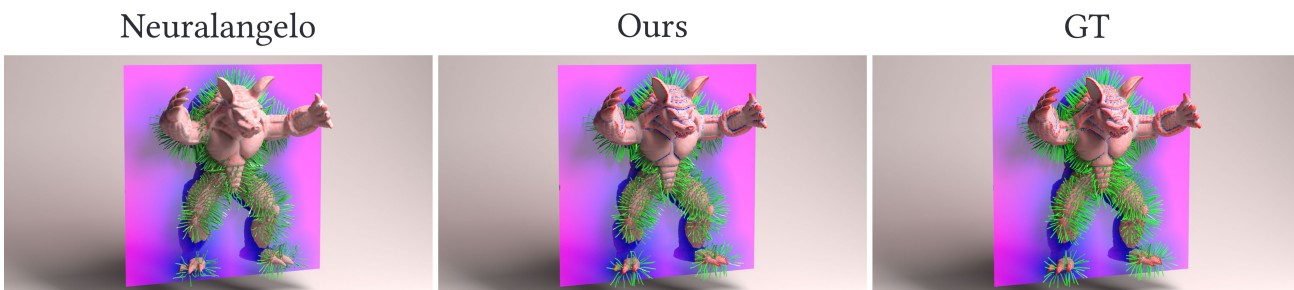

*Figure 23.* **SDF Curvature (Armadillo).** Mesh colored by mean curvature, green lines show gradient direction. Hermite-NGP achieves gradient MAE 0.0478 vs. NeuralAngelo's 0.1009 (2.1× lower).

| Neuralangelo | Ours | GT |
|---|---|---|

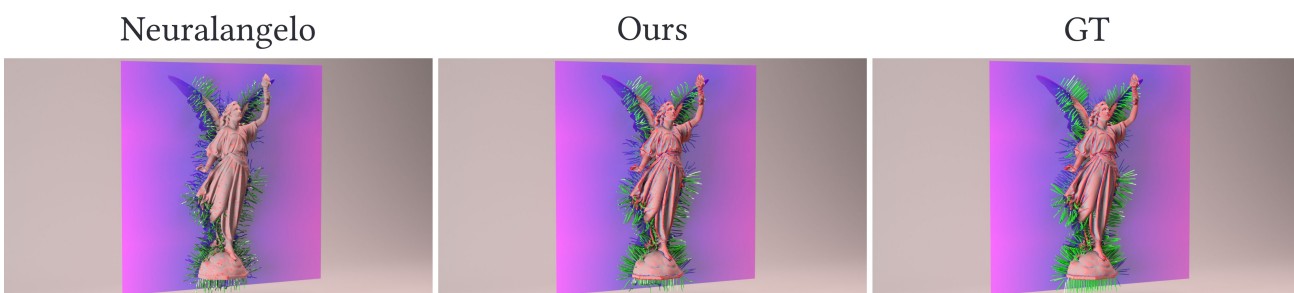

*Figure 24.* **SDF Curvature (Lucy).** Mesh colored by mean curvature, green lines show gradient direction. Hermite-NGP achieves gradient MAE 0.0418 vs. NeuralAngelo's 0.1213 (2.9× lower).

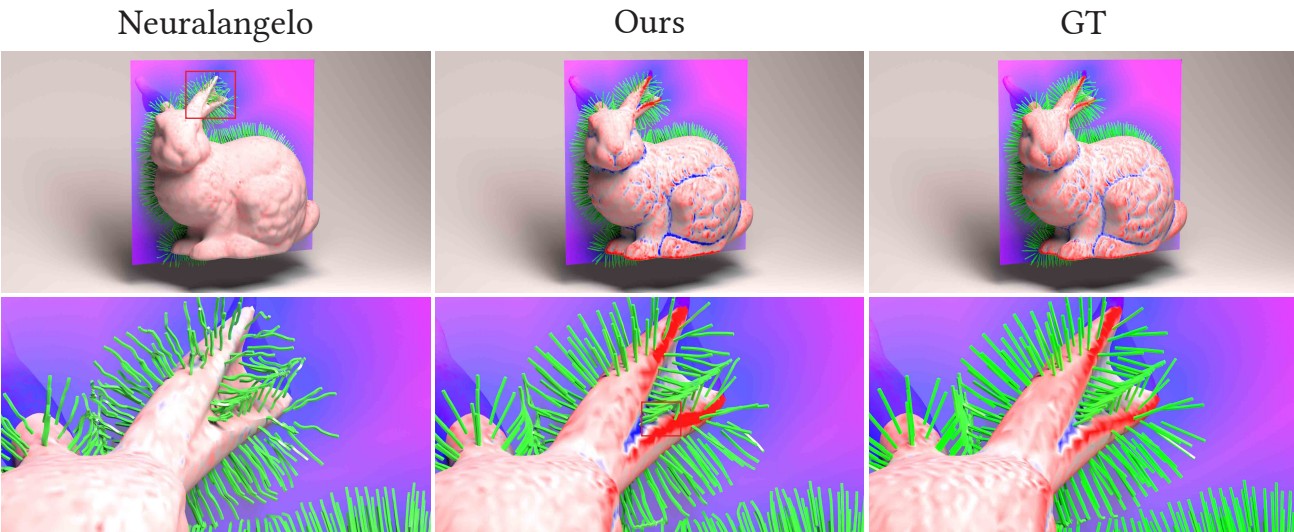

| Neuralangelo | Ours | GT |
|---|---|---|

*Figure 25.* **SDF Curvature (Fandisk).** Mesh colored by mean curvature, green lines show gradient direction. Hermite-NGP achieves gradient MAE 0.0516 vs. NeuralAngelo's 0.1064 (2.1× lower), preserving sharp edges.

| Neuralangelo | Ours | GT |
|---|---|---|

*Figure 26.* **SDF Curvature (Bunny).** Mesh colored by mean curvature, green lines show gradient direction. Hermite-NGP achieves gradient MAE 0.0416 vs. NeuralAngelo's 0.0887 (2.1× lower).

