# OpenReview forum: "Hermite-NGP: Gradient-Augmented Hash Encoding for Learning PDEs"
_ICML.cc/2026/Conference — ICML 2026 regular_

### Official Review · Reviewer_wdxf · 2026-03-10

**Soundness:** 2
**Presentation:** 2
**Significance:** 2
**Originality:** 2
**Overall Recommendation:** 5
**Confidence:** 5

**Summary:**

Hermite-NGP aims to avoid the instability and cost of relying on autograd or finite differences for higher-order derivatives in standard NGP-style approaches. The method is applied within a PINN framework and evaluated on multiple 2D/3D PDEs, complex geometries, and geometric differential operator tasks, where the paper reports improvements in error, convergence, and training efficiency over several existing methods.

**Compliance With Llm Reviewing Policy:**

Affirmed.

**Final Justification:**

Thanks for the response. I also reviewed the authors’ replies to the other reviewers. Overall, the rebuttal is largely convincing: it provides new evidence, additional quantitative support, a more clear clarification of the novelty and the concerns regarding the baselines. The authors also maintained a highly professional and constructive tone throughout their response. As a result, my concerns have been addressed, and I am raising my score.

**Key Questions For Authors:**

Major concerns

[1] Hermite-NGP is essentially a combination of "Hermite interpolation, hash encoding, and explicit storage of derivative coefficients". The main trick is to store derivatives in the representation itself instead of relying on autograd or finite differences. This is not especially groundbreaking.

[2] The paper repeatedly suggests that its analytic second-order derivatives fundamentally solve the limitations of standard NGP-style methods, and even hints that finite differences face an intrinsic accuracy ceiling. But this is not argued carefully enough. The paper does not convincingly show that Hermite-NGP would still enjoy such a dramatic advantage against stronger alternatives with smoother representations, better encodings, or more thoughtful numerical differentiation schemes. Part of the argument feels like beating up weak baselines and calling it a fundamental victory.

[3] The paper repeatedly highlights dramatic gains like “10×,” “100×,” or being the “only” method that converges, but the comparison set is underwhelming. Some baselines are dated, some simply fail, and some results are taken from original papers instead of being fully rerun under a unified setup. Under these conditions, the huge gains are hard to fully trust. It is entirely possible that part of the apparent superiority comes from weak or uneven baselines rather than from a truly dominant method.

[4] The paper tries hard to present the method as a broadly applicable solver for many 2D/3D PDEs, complex geometries, and geometric differential operators. However, the evidence still comes mainly from a collection of fairly standard PINN-style benchmarks. There is no serious demonstration on high-dimensional problems, strongly coupled multiphysics settings, challenging time-dependent PDE regimes, or meaningful comparisons with mature numerical PDE solvers. Calling this a “general” solver feels far too ambitious and, honestly, a bit reckless.

[5] Hermite-NGP obtains analytic derivatives by explicitly storing mixed partial derivatives, which inevitably introduces additional memory and representation overhead. That cost can blow up quickly as dimensionality increases. The paper mentions this only briefly, almost as an afterthought, but this is not a minor caveat at all. It is a central limitation that directly affects whether the method can scale beyond the relatively friendly settings shown in the paper.

Minor concerns

[1] Phrases like “order-of-magnitude lower error,” “general solver,” and “exact differential operators” are used very boldly throughout. The tone often feels more like marketing than careful scientific argument.

[2] The main metrics are relative $L^2$ error, training time, and memory. Those are useful, but they are not enough to establish that the method is truly more reliable from a numerical analysis perspective. Stronger evidence on stability, convergence behavior, error sources, and physical consistency would make the claims much more convincing.

[3] The paper only briefly mentions storage overhead, higher-dimensional scaling, and activation-function dependence, but these are not small issues. In particular, the dependence on SIREN-style representations, the scalability of storing derivatives, and the effect of hash collisions on higher-order derivatives all deserve a much more honest and detailed discussion.

**Limitations:**

yes

**Strengths And Weaknesses:**

Strengths: The paper identifies a clear and important problem: high-order derivative computation in neural PDE solvers is often both expensive and unstable, especially in PINN-style methods and geometry-aware tasks.

Weaknesses: See the part "Key Questions For Authors" below.

---

> ### Author Rebuttal · Authors · 2026-03-31
>
> We thank the reviewer for the detailed critique. We address each concern below.
>
> **[Q1] Novelty.**
> The main novelty of our approach is the explicit separation of function and gradient representations within the NGP architecture, which addresses a key limitation of existing NGP methods in neural PDE applications where accurate gradients are essential. In particular, naive gradient representation (i.e., auto-diff) prevents standard NGP from being effectively applied to neural PDEs, thereby limiting the ability to leverage its strengths in representing multi-resolution fields for PDE problems with complex geometries.
>
> On the other hand, while gradient-augmented representations have been well studied in classical numerical methods (e.g., on grids [Nave et al., 2010] or particles [Jiang et al., 2015]) for high-order accuracy, they have not been explored in neural field representations with multi-resolution hash encoding. To our knowledge, Hermite NGP is the first to incorporate gradient augmentation into NGP, improving gradient accuracy while preserving hash-based efficiency.
>
> **[Q2] Advantage over stronger alternatives not shown.**
> Our focus is on neural PDE, rather than improving classical numerical solvers. Accordingly, we primarily compare against strong PINN and neural representation baselines (e.g., PirateNet, PIG, NeuralAngelo, among others), following standard practice in this domain. Comparisons to classical solvers are not our main focus due to the differing problem objectives, and we will clarify it more explicitly in the paper to avoid potential confusion.
>
> We additionally included the recent $\partial^\infty$-Grid baseline (Kairanda et al., 2026, ICLR) and other requested comparisons on Helmholtz $a=10$:
>
> | Method | L2 Error |
> |--------|----------|
> | **Hermite-NGP** | **3.35e-5** |
> | $\partial^\infty$-Grid | 6.07e-3 |
> | SIREN | 6.67e-2 |
>
> We also evaluated Hermite-NGP on ∂∞-Grid’s image reconstruction benchmark (camera 256×256, with gradient/Laplacian supervision):
>
> | Task | Hermite-NGP | ∂∞-Grid | SIREN | K-Planes |
> |------|-------------|---------|-------|----------|
> | Gradient PSNR | **32.45** | 32.24 | 32.12 | 17.96 |
> | Laplacian PSNR | **13.28** | 12.19 | 11.82 | — |
>
> These results suggest Hermite-NGP is competitive for both PDE fitting and signal reconstruction. We welcome specific suggestions for additional baselines.
>
> **[Q3] Baselines rerun**
> Most baselines were rerun under the same unified setup for fair comparison. For Flow Mixing, SPINN ($2.90 \times 10^{-3}$) and PIG ($2.67 \times 10^{-4}$) were originally from their own settings, we have rerun both and confirm the same conclusion. "Fail" denotes genuine non-convergence (residual >1.0), not inconsistent settings.
>
> **[Q4] Scope and tone.**
> In the revision, we will follow the suggestion and make the scope more precise, while improving the exposition to be more quantitative and restrained. For example, we will replace qualitative claims such as “order-of-magnitude” with specific experimental numbers, remove terms like “general solver” (L064, L430), and revise “exact” to “analytic,” among other adjustments.
>
> **[Q5] Storage overhead**
> We provided analysis in Appendix C.4; in the revision, we will move the analysis to the main text and expand the discussion. FD-based Hash methods(INGP-FD) also incur substantial memory overhead: central differences in 3D require \(2d+1=7\) forward passes per iteration, each requiring memory for backpropagation. We additionally profiled 3D training memory on Helmholtz 3D (a=10, 40k collocation points):
> | Ours Params | Ours Mem | FD Params | FD Mem |
> |----------------|-------------|-----------|--------|
> | 150K           | 957 MB      | 117K      | 1402 MB |
> | 2.12M          | 988 MB      | 314K      | 1405 MB |
> | 8.41M          | 1084 MB     | 4.77M     | 1502 MB |
> | 33.6M          | 1465 MB     | 21.0M     | 1783 MB |
>
> In experiments, Hermite-NGP uses lower peak GPU memory, since FD requires 7 forward passes and stores substantially more intermediate activations.
>
> **[Q6] Insufficient metrics.**
> We reported relative error, training time, memory, convergence curves, and ablations over hash table capacity and resolution levels. To further address this concern, We have now run 5-seed experiments (mean ±\pm ± std):
> | Experiment | L2 Error |
> |:---|:---|
> | Helm. 2D ($a=10$) | $3.35e{-5} \pm 4.4e{-6}$ |
> | Helm. 2D ($a=20$) | $1.07e{-4} \pm 3.64e{-5}$ |
> | Helm. 2D ($a=100$) | $7.34e{-2} \pm 6.06e{-3}$ |
> | Helm. 3D ($a=10$) | $7.29e{-3} \pm 1.10e{-3}$ |
> | TG vortex | $5.69e{-5} \pm 3.7e{-6}$ |
> | Advection ($c=30$) | $9.47e{-5} \pm 2.5e{-5}$ |
>
> These results indicate stable training across tested settings. We will include these statistics in the revision.
>
> **[Q7] Limitations discussion insufficient.**
> We will move more of the hash collision and storage scaling ablations (Appendix C.2–C.3) into the main text; clarify in Sec. 5 that the SIREN activation is decoupled from the Hermite hashing stage.

---

> > ### Author Rebuttal · Reviewer_wdxf · 2026-04-03
> >
> > Thanks for the response. I also reviewed the authors’ replies to the other reviewers. Overall, the rebuttal is largely convincing: it provides new evidence, additional quantitative support, a more clear clarification of the novelty and the concerns regarding the baselines. The authors also maintained a highly professional and constructive tone throughout their response. As a result, my concerns have been addressed, and I am raising my score.

---

> > > ### Author Response · Authors · 2026-04-04
> > >
> > > Thank you for the acknowledgement and your helpful feedback. We also sincerely appreciate your thoughtful follow-up and your decision to raise the score! We will reflect these clarifications and added evidence in the revision.

---

### Official Review · Reviewer_NHLu · 2026-03-13

**Soundness:** 4
**Presentation:** 3
**Significance:** 3
**Originality:** 4
**Overall Recommendation:** 5
**Confidence:** 4

**Summary:**

Hermite-NGP explicitly stores function values and mixed partial derivatives at hash grid vertices, constructing a C^1 continuous spatial field within grid cells through Hermite interpolation. This method aims to improve the speed and accuracy of spatial derivative computation in neural partial differential equation (PDE) solvers. The paper also introduces a multi-resolution coarse-to-fine curriculum training strategy, analogous to multigrid methods, to accelerate the optimization process. Across a variety of 2D and 3D PDE benchmarks, Hermite-NGP achieves a relative L^2 error that is an order of magnitude lower than previous neural PDE methods. Simultaneously, it reduces the wall-clock convergence time by 2 to 10 times, with per-epoch training times as low as 3.5 milliseconds.

**Compliance With Llm Reviewing Policy:**

Affirmed.

**Key Questions For Authors:**

Since each query during the interpolation process needs to locate 2^d vertices, and each vertex requires reading all 2^d coefficients, a single grid cell in the 3D case requires reading 64 feature values. Although the total number of parameters can be controlled by adjusting the hash table size, will the random memory access bandwidth become the core bottleneck for further accelerating the algorithm? Although the authors briefly mention the bandwidth limitation under the 2^18 table size in Appendix C.4, it is recommended to add an in-depth discussion in the main text regarding the memory access pressure when scaling to dimensions d >= 3.

**Limitations:**

yes

**Strengths And Weaknesses:**

Strengths:
1. Explicitly storing derivative information breaks the reliance on finite differences and automatic differentiation in neural PDE solving. The model can accurately calculate gradients, Jacobians, and Hessians directly through the mathematical formulas of the basis functions in a single forward pass, balancing both speed and numerical precision.
2. Drawing inspiration from multigrid V-cycles in traditional numerical computation, the authors designed a curriculum learning strategy that progressively activates hash grid resolutions from low to high.
3. The paper conducts a comprehensive evaluation across multiple benchmarks covering elliptic, hyperbolic, and time-dependent equations. In the most challenging high-frequency Helmholtz (a=100) test, Hermite-NGP is the only method capable of converging.
4. The paper provides a detailed ablation analysis of hash table capacity allocation and draws the highly insightful empirical conclusion that the first-derivative table is the most sensitive to hash collisions, providing valuable reference for future work.



Weaknesses:
1. Explicitly parameterizing mixed partial derivatives introduces additional storage overhead, and this overhead increases sharply in higher dimensions (d > 3), which limits its potential for scaling to high-dimensional PDEs.
2. In the pursuit of extreme derivative computation speed, the current MLP analytic derivative propagation mechanism relies deeply on the mathematical properties of the second derivatives of specific activation functions (such as SIREN). This means that without modifying the underlying theoretical derivations, it is difficult for the model to flexibly swap to other types of activation functions, and currently it only supports up to second-order differential operators.

---

> ### Author Rebuttal · Authors · 2026-03-31
>
> We sincerely thank the reviewer for the insightful evaluation and constructive suggestions.
>
>
> **[W1/Q1] Storage overhead and memory scaling.**
>
> We agree that storing \(2^d\) derivative channels per vertex is a real cost, and that this limitation deserves clearer discussion. This overhead grows with dimension, although our target setting is spatial PDEs with $d \le 4$(e.g., 3D + time), which covers most applications we study.
>
> At the same time, FD-based methods also incur substantial memory overhead in higher dimensions: central differences in 3D require \(2d+1=7\) forward passes per iteration, each storing activations for backpropagation. We additionally profiled 3D training memory on Helmholtz 3D (\(a=10\), 40k collocation points):
>
> | Hermite Params | Hermite Mem | FD Params | FD Mem |
> |----------------|-------------|-----------|--------|
> | 150K           | 957 MB      | 117K      | 1402 MB |
> | 2.12M          | 988 MB      | 314K      | 1405 MB |
> | 8.41M          | 1084 MB     | 4.77M     | 1502 MB |
> | 33.6M          | 1465 MB     | 21.0M     | 1783 MB |
>
> In these 3D settings, Hermite-NGP uses lower peak memory despite storing \(2^3=8\) derivative channels per vertex, because FD requires seven forward passes and stores substantially more intermediate activations. We will move this analysis from Appendix C.4 to the main text and expand the limitations discussion in the revision.
>
>
>
> **[W2] Activation function dependence.**
>
> We would like to clarify that the analytic MLP derivative propagation is not restricted to SIREN,  it works with any twice-differentiable activation function (e.g., Swish, softplus, GeLU). The hash encoding derivatives are entirely activation-independent. Furthermore, one can also fall back to automatic differentiation for the MLP component at the cost of speed, the analytic derivatives from the hash encoding stage are always preserved, which is the primary source of our speedup. We are happy to provide experiments with additional activation functions and corresponding CUDA kernels if the reviewer finds this valuable.

---

> > ### Author Rebuttal · Reviewer_NHLu · 2026-04-03
> >
> > The authors have addressed or committed to address my concerns.

---

> > > ### Author Response · Authors · 2026-04-04
> > >
> > > Thank you for the acknowledgement and helpful feedback. We will reflect these clarifications in the revision.

---

### Official Review · Reviewer_HjTL · 2026-03-14

**Soundness:** 4
**Presentation:** 4
**Significance:** 4
**Originality:** 4
**Overall Recommendation:** 5
**Confidence:** 4

**Summary:**

This paper proposes Hermite-NGP, a multi-resolution hash encoding for neural PDE solving that stores not only function values but also partial derivatives at grid vertices, and reconstructs the field with Hermite interpolation. The authors show that this can enable one to construct a C1 continuous representation and analytic first and second order derivatives.The authors show that their method works empirically over a very broad set of PDEs, esp showing strong results on high-frequency Helmholtz equation.

**Compliance With Llm Reviewing Policy:**

Affirmed.

**Key Questions For Authors:**

Can you report multi-seed means and standard deviations for the main benchmarks, esp for high frequency Helmholtz equations? This would help confirm that the gains are robust and not driven by a few favorable runs.

It would be very interesting to see what the architectural tradeoffs with the neural network here would be. The authors have used relatively shallow networks and are still getting good performance, but what happens as the depth is further increased. I am assuming it would result in a tradeoff since the memory footprint of the method would also increase, however given the stability issues with PINN style works, its an interesting thing to study.

**Limitations:**

Addressed in previous sections.

**Strengths And Weaknesses:**

I think this is a strong paper with a clean idea and comprehensive experiments. The paper is very well written. The authors show that their methodology works on a wide range of real world (not synthetic) PD datasets that are very different in their characteristics. There are also ablations that help isolate where the improvements are coming from. The authors also show results on 3D datasets (such as 3D helmholtz) which is very convincing.
Their methodology also consistently outperforms the baselines across multiple families of PDEs.

---

> ### Author Rebuttal · Authors · 2026-03-31
>
> We thank the reviewer for the thoughtful questions and helpful feedback.
>
> **[Q1] Multi-seed means and standard deviations.**
>
> We have conducted 5-seed experiments across all main benchmarks including **high frequency** Helmholtz:
> | Experiment | L2 Error |
> |:---|:---|
> | Helm. 2D ($a=10$) | $3.35e{-5} \pm 4.4e{-6}$ |
> | Helm. 2D ($a=20$) | $1.07e{-4} \pm 3.64e{-5}$ |
> | Helm. 2D ($a=100$) | $7.34e{-2} \pm 6.06e{-3}$ |
> | Helm. 3D ($a=10$) | $7.29e{-3} \pm 1.10e{-3}$ |
> | TG vortex | $5.69e{-5} \pm 3.7e{-6}$ |
> | Advection ($c=30$) | $9.47e{-5} \pm 2.5e{-5}$ |
>
> The low relative standard deviations (<15%) confirm that the gains are robust and not driven by favorable initializations.
>
> **[Q2] Network depth architectural tradeoffs.**
>
> We ablated MLP depth on 2D Helmholtz ($a=20$) with $w=128$ and a smaller hash setting $(12/12/10)$ to better isolate the effect of MLP depth.
>
> | Depth | Best L2   | ms/epoch |
> |-------|-----------|----------|
> | $1$     | $1.02e{-3}$   | $2.17$     |
> | $2$    | $6.81e{-4}$   | $3.07$     |
> | $3$     | $4.27e{-4}$   | $3.95$     |
> | $4$     | $4.03e{-4}$   | $4.86$     |
>
> Deeper networks improve accuracy (d=4 is 1.7x better than d=2) but each layer adds ~0.9 ms/epoch. Our default (d=2) balances accuracy and speed. The hash encoding carries most representational burden, the MLP is a lightweight decoder with diminishing returns beyond d=2.
>
> We will include all of these experiments in the revised paper.

---

> > ### Author Rebuttal · Reviewer_HjTL · 2026-04-03
> >
> > I thank the authors for their rebuttal and some extra experiments related to the depth of the networks.

---

> > > ### Author Response · Authors · 2026-04-04
> > >
> > > Thank you for the acknowledgement and for taking the time to review the additional experiments. We appreciate your thoughtful feedback, and we will include these results and clarifications in the revised paper.

---

### Official Review · Reviewer_bbzf · 2026-03-19

**Soundness:** 3
**Presentation:** 2
**Significance:** 4
**Originality:** 3
**Overall Recommendation:** 4
**Confidence:** 4

**Summary:**

The paper presents a neural PDE solver inspired by Instant Neural Graphics Primitives (I-NGP). The work augments the (non-differentiable) hash grid-based encoding of I-NGP with gradients, and it supports analytical calculation of exact differential operators up to second order. The work is evaluated for solving a variety of PDEs in 2D, 3D (including temporal ones). The method is shown to be faster and more accurate than prior work on PINN solvers and the simple finite-difference-based extension to I-NGP.

**Compliance With Llm Reviewing Policy:**

Affirmed.

**Final Justification:**

Thanks for the rebuttal. It addresses all of my concerns, and I recommend acceptance. Please carefully revise the factual errors in the writing of I-NGP variants.

**Key Questions For Authors:**

1.	The comparison to higher-order, for example, cubic I-NGP baseline. Why do they fail? Please explain.
2.	Could you compare to Siren? This is a relevant baseline and might give higher accuracy than some of the compared PINNs. Moreover, a recent work [Kairanda et al. ICLR 2026] also introduced a neural PDE solver using differentiable feature grids. It should be cited, and a comparison would also be valuable.
3.	How do you impose boundary conditions that require derivatives, such as Neumann? What adjustments are required?
4.	In Section 7.4, there is an experiment of solving the 3D Poisson with mesh boundary conditions. While the numerical results are provided, could you please include the visualization of the results?
5.	The hash function in general is not differentiable. How does Hermite-NGP account for this?

**Limitations:**

yes

**Strengths And Weaknesses:**

Soundness:

++ The proposed work is technically sound. The idea of storing partial derivatives at grid values is quite practical. Hermite interpolation, hashing, and the computation of analytic derivatives are a great strategy.

++ The claims are evaluated in a comprehensive way, across dimensions, PDEs, geometries, and compared to multiple baselines.

Presentation:

++ The paper is mostly well-structured and well-written, and easy to read and understand.

-- Although well-written, I noticed a lot of factual errors in the draft. For example, the opening line of the introduction, “Multi-resolution hash encoding, exemplified by Instant Neural Graphics Primitives (I-NGP) (Mu ̈ller et al., 2022), along with its many variations (e.g., see (Fridovich-Keil et al., 2022; Barron et al., 2023; Kerbl et al., 2023; FridovichKeil et al., 2023; Chen et al., 2022; Cao & Johnson, 2023; Kim et al., 2024; Chen et al., 2025; Zou et al., 2024)), …” is highly incorrect.  The cited works are not variations of I-NGP, in fact some of the works appeared even before I-NGP. Rather, all the works fall in the category of “efficient neural scene representation, with the efficiency coming from combining classical data structures (grid, tree, Gaussians) with neural counterparts. I-NGP is an example in the category, and not the predecessor.  Similarly, the related work section, 3D Gaussian Splatting is not a grid based method.

-- The papers mention Hermite-NGP is a “C1-continuous spatial field within each grid cell, admitting well-defined and analytic second-order derivatives everywhere” (L086). I think the authors instead mean C2 continuous. Note that C1 continuity implies first derivatives exist, but doesn't say anything about second derivatives. C2 continuity means second derivatives exist.

-- Moreover, it is unclear what Fig. 3 is; it is explained well, and I don’t get it. NeuralAngelo performs surface reconstruction from RGB images and is not designed for solving Eikonal PDEs. How is it used exactly for comparison?

-- Minor comments on writing
1.	In line 015, the full form of NGP, ie, neural graphics primitives, is required.
2.	L170, this should be N_l() instead of C_l().
3.	Equation 2 and the specific interval [0,1] (at L147) are inconsistent.
4.	In L149, note that d-linear interpolation also precludes analytic evaluation of PDE operators involving first-order derivatives (as the first-order derivatives don’t exist at cell boundaries).

Significance:

++ Solving PDEs is an important problem across engineering and graphics, thus the address problem is significant. I guess the method should also be able to represent signals similar to I-NGP. Could you include some results?

Originality:

++ The method is quite novel, borrowing interesting ideas from gradient-augmented field representants and has encoding.

---

> ### Author Rebuttal · Authors · 2026-03-31
>
> We thank the reviewer for the detailed and constructive feedback. We address each concern below.
>
> **[W1] Factual errors in introduction framing.**
>
> We agree and will correct: the cited works are from the broader neural representation literature, not "variations of I-NGP," and 3DGS is not grid-based. We will revise the introduction and related work accordingly.
>
> **[W2] C1 vs. C2 continuity (L086).**
>
> We thank the reviewer for catching this. Cubic Hermite interpolation yields a $C^1$ field: function values and first derivatives are continuous across cell boundaries, while second derivatives are piecewise analytic within each cell but may be discontinuous across boundaries, so the field is not globally $C^2$. In practice, after the hash encoding produces piecewise-analytic second derivatives, the subsequent MLP and PDE-loss optimization lead to smooth and accurate second-order behavior across cell boundaries. Higher-order schemes such as quintic Hermite could provide stronger continuity, but at substantially higher parameter cost. Cubic Hermite is a practical tradeoff. We will add this discussion to the paper.
>
> **[W3] Fig. 3**
>
> For Fig. 3, our goal is to illustrate the benefit of gradient-augmented interpolation for implicit surface representation. Even when all four corner values in a cell are positive, adding gradient information allows the cell to represent a more complex local interface. Hermite interpolation is one such gradient-augmented scheme, and we use it here to illustrate this point. We will explain this more clearly in the paper.
>
> **[W4] NeuralAngelo comparison.**
>
> Since NeuralAngelo uses an FD-based derivative design similar in spirit to INGP-FD, we use its architecture here as an FD-based baseline under the same SDF + gradient fitting objective. The comparison is therefore about derivative/representation quality (as shown in the figure through better gradients and curvature), not its original RGB -> surface reconstruction task. We will clarify this in the text and caption.
>
> **[W5] Minor writing errors.**
>
> We acknowledge all four issues and will fix them in the revision. We note that L149 (d-linear lacks first-order derivative continuity) further motivates the need for Hermite interpolation.
>
> **[Q1] Cubic I-NGP baseline — why does it fail?**
>
> Cubic interpolation on standard NGP still relies on finite differences or automatic differentiation through the hash-encoded features to obtain derivatives. Hash collisions inject high-frequency noise into the stored feature values, and computing derivatives via FD on these noisy features amplifies the noise. In contrast, Hermite-NGP explicitly stores and learns the derivative field as a first-class optimization target, distributing the representation burden across function and derivative channels. This makes the encoding structure more stable and avoids the noise amplification that degrades cubic I-NGP.
>
> **[Q2] SIREN and Kairanda et al. (ICLR 2026).**
>
> We ran comparisons with both methods. We tested their method on our PDE task set (Helmholtz, a=10).
>
> | Method | L2 Error |
> |--------|----------|
> | Hermite-NGP (ours) | 3.35e-5 |
> | ∂∞-Grid (Kairanda et al.) | 6.07e-3 |
> |SIREN | 6.67e-2 |
>
> We also test Hermite-NGP on ∂∞-Grid’s image reconstruction benchmarks (camera 256×256, with gradient and Laplacian supervision), which evaluate its ability to represent signals.
>
> | Task | Hermite-NGP | ∂∞-Grid | SIREN | K-Planes |
> |------|-------------|---------|-------|----------|
> | Gradient PSNR | **32.45** | 32.24 | 32.12 | 17.96 |
> | Laplacian PSNR | **13.28** | 12.19 | 11.82 | — |
>
> Hermite-NGP outperforms on both PDE and signal tasks. The key difference: ∂∞-Grid uses RBF interpolation on co-located grids; we use Hermite interpolation with explicit derivative storage on hash-encoded grids.
> For completeness, our results use the default setup in the paper: a 2-layer 128-width MLP with 8 hash levels (scale $=2.0$). For ∂∞-Grid, we used their original code with a larger grid(256) and longer training ($100$k iterations).  We will include these settings in the revised paper.
>
> **[Q3] Neumann boundary conditions.**
>
> Neumann BCs are imposed via soft loss on boundary collocation points, same as standard PINNs.
>
> **[Q4] 3D Poisson mesh visualization.**
>
> The ground truth in the figure is obtained from a finite-element discretization at 256^3 resolution, with the resulting linear system solved using SciPy. We will clarify this in the caption.
>
> **[Q5] Hash function differentiability.**
>
> The hash function maps integer grid coordinates to table indices, it is a discrete index lookup and is not part of the continuous computation graph. It determines which coefficients to read from the hash table, but spatial derivatives are computed entirely through the Hermite basis functions, which are smooth polynomials of the continuous query position. This is the same principle as in standard I-NGP (where the hash is also discrete).

---

> > ### Author Rebuttal · Reviewer_bbzf · 2026-04-03
> >
> > Thanks for the rebuttal. It addresses all of my concerns, and I recommend acceptance. Please carefully revise the factual errors in the writing of I-NGP variants.

---

> > > ### Author Response · Authors · 2026-04-04
> > >
> > > Thank you for the acknowledgement and positive recommendation. We appreciate your careful reading and helpful feedback, and we will make sure to correct the factual issues around the discussion of I-NGP variants in the revision.

---

### Decision · Program_Chairs · 2026-04-30

**Decision:**

Accept (regular)

**Comment:**

This paper tackles an important problem in neural PDE solving, i.e., how to circumvent the limitations/inefficiencies in PDE residual computations due to the intrinsic problems with automatic differentiation and finite differences. Its proposed solution is neat: store derivatives at points and use Hermite interpolants to analytically compute gradients. This simple idea is implemented effectively and the authors compare their method to a number of PINN variant baselines on a set of 2- and 3-D experiments. The reviewers uniformly appreciated the contributions and the rebuttal discussions. The paper should be accepted but in a camera-ready version, the authors should clearly discuss the limitations of their method namely i) it is unclear how this formulation translates to complex (non-cartesian) domains ii) the method is based on a strong form of the residual where as the weak form is often necessary for mathematical correctness iii) the relevant comparison is not with PINN type methods but rather with numerical PDE solvers such as FDM or FEM. The authors provide no such comparison, particularly in terms of runtimes as well as accuracy. This will provide a balanced perspective to the readers and usher further adoption.